# DE-BIASING DIFFUSION: DATA-FREE FP8 QUANTIZATION OF TEXT-TO-IMAGE MODELS WITH BILLIONS OF PARAMETERS

## ABSTRACT

Diffusion neural networks have become the go-to solution for tasks involving automatic image generation, but the generation process is expensive in terms of memory, energy, and computational cost. Several works have aimed to reduce the cost via quantization to 8 bits or less. Despite that, half-precision (FP16) is still the default mode for large text-to-image generation models — where reducing numerical precision becomes more challenging. In this work, we show that the reduction of quantization bias can be more important than the reduction in quantization (mean square) error. We propose a data-free method for model quantization, with the goal of producing images that are indistinguishable from those generated by large, full-precision diffusion models. We show that simple methods like stochastic rounding can decrease the quantization bias and improve image generation quality with little to no cost. To close the remaining gap between full-precision and quantized models, we suggest a feasible method for partial stochastic-rounding of weights. When using the MS-COCO dataset as the baseline, we show our quantization methods achieve as good FID scores as the full-precision model. Moreover, our methods decrease the quantization-induced distortion of the images generated by the full-precision model, with the distortion decreasing with the number of diffusion steps.

## 1 INTRODUCTION

At this time, the best automatic method to generate high-quality images is using diffusion models (Sohl-Dickstein et al., 2015) with a deep neural network (DNN) architecture. While the quality of the images generated by diffusion models is currently unmatched (Dhariwal & Nichol, 2021), these models have a high computational cost. Diffusion processes are inherently iterative and require multiple steps for the generation of each output image. When studying the energy cost of different AI models, Luccioni et al. (2023) claimed that generating a single image with the smallest text-to-image model tested has the same energy cost as fully charging a smartphone, and the process emits 10 times more CO2 than a text-to-text process performed by the largest language model in that study. Furthermore, diffusion models have a high memory cost, which scales with the image resolution. This is a major limiting factor when using a memory-bounded accelerator to process these models.

While the energy requirements are higher than ever before, the challenge of increasing efficiency in DNNs, by itself, is not new. The most common solution, that spans most DNN applications, is DNN quantization (e.g., Gupta et al. (2015); Hubara et al. (2017)). DNNs that use datatypes with lower numerical precision require less memory and can be computed much more efficiently on dedicated hardware (van Baalen et al., 2023). The downside of using quantization is that lower precision datatypes (with fewer bits) result in less accurate models. Reducing the quantization inaccuracies while minimizing quantization data size is the focus of most research on quantization in DNNs.

Several prior works (Li et al., 2023; Yang et al., 2023; He et al., 2024; Shen et al., 2023) have tackled the issue of quantization in diffusion models, and have suggested distinctive methods to quantize these models post-training, without harming the model's output to a significant degree. The apparent success of quantizing latent diffusion models, did not, however, extend to standard practice. Despite the high demand for lighter, faster, SOTA text-to-image models, the standard workflow for latent

diffusion models still relies heavily on half-precision datatypes (16-bit), or naive FP8 quantization, which is publicly distributed for Flux. As we explain next, in Sec. 2, existing methods struggle to scale to larger models (with more than a billion parameters), such as SDXL and Flux, and offer limited functionality that hampers their wide-scale adoption.

In this work, we aim to bridge the gap and suggest 8-bit diffusion models that are more relevant to practitioners. Our work will focus on the popular open-sourced Stable Diffusion XL and Flux models (Podell et al., 2023; Labs, 2024), and will aim to generate images that look just like the images produced by the full-precision, unquantized models. We will do so without relying on any external data, as it is not, in general, broadly available in the context of SOTA image generation applications. To accomplish this goal, we will leverage the fact that the diffusion process includes multiple iterative steps. Consequently, the effect of quantization noise is dominated by its bias. As we will show, reducing this bias in data-free settings is possible and beneficial for all large models: from the smaller SD1.4 to Stable-Diffusion XL and Flux.

## 2 RELATED WORKS

Several existing works proposed varying schemes for the quantization of diffusion models. In Li et al. (2023), the authors suggest that by accounting for the changes made to the statistics of the quantized neural network during the diffusion process, it is possible to quantize diffusion models (post-training) to INT4/INT8 data types, which reduces the computational cost and memory substantially. The paper shows that despite the aggressive quantization, the unconditional models manage to maintain the high quality of generated images. Conditional Text-To-Image models were also examined and were shown to be of high quality, despite some degradation when compared with the non-quantized models.

In Yang et al. (2023), the authors propose integer quantization of latent-diffusion models as well, which they achieve by analyzing the effects of different components in the U-Net architecture. With a single-calibration step, intended to optimize the layer-level signal-to-quantization-noise-ratio (SQNR), the authors use quantization corrections to transform full-precision, pre-trained models into partially quantized models. They show that their quantized model produces high-quality images. Specifically, when applied over high-quality stable diffusion models (SD 1.5 (Rombach et al., 2022) or SDXL (Podell et al., 2023)), the quality was assessed to be on par with non-quantized diffusion models.

Diffusion models were also quantized to integers in He et al. (2024) (PTDQ). There, the authors suggest a correction to the DDPM scheduling regime, which they denote as *Variance Schedule Calibration* (VSC). When used in combination with step-aware mixed precision, their method achieves good metrics for images generated over LSUN datasets. In addition to the previous methods, the best results in the paper rely on bias correction on the models' output, which is adjusted per timestep. For conditional diffusion models, the bias correction is also calibrated per class.

Post-training quantization methods for FP8 datatypes were reviewed in Shen et al. (2023). Unlike the previous papers that targeted diffusion models, Shen et al. (2023) covered a vast amount of tasks, studying the effect of different data types on the quality of different neural networks. When applied to stable diffusion models, the paper shows that the FP8-M4E3 format (4 Mantissa bits) is superior to other formats (FP8 or INT8). The results were further improved by implementing dynamic range calibration, where values were scaled based on the minimum and maximum over training samples.

The most noteworthy practical implementation of quantized diffusion models is offered by Nvidia's TensorRT (Cheng et al., 2024), which is associated with Li et al. (2023). In a blog post, the authors report results for either FP8 or INT8 quantization, arguing that the performance of FP8 is preferable since INT8 quantization can not be extended to all the operations in the attention block. While Cheng et al. (2024) did not include any quality metrics to numerically assess the generated images, it did establish the similarity between images generated by the full-precision and the quantized models to be a key metric, and was the only work that offered official implementation for the quantization of Stable-Diffusion XL.

The primary criterion for assessing image quality in each of the academic works mentioned in this section was the Frechet Inception Distance (FID). In the case of unconditional image generation, the images generated by all methods appear to be very similar to the baseline images, even when using a very low quantization format (4 bits). However, scaling these methods to large, text-to-image models is more challenging: Even when using mixed precision (excluding the Softmax layers), the

images generated when using these methods (Li et al., 2023; Yang et al., 2023) diverge from the full-precision image. Furthermore, as we show in Appendix A, the existing methods have multiple limitations, that may exclude them from being utilized by most users. For example, we show that scaling TensorRT to SDXL results in lower image quality (higher FID) and that the solutions are not applicable when the number of diffusion steps reaches 50 (The default number of steps for SDXL).

In Appendix D, we provide a breakdown of all the primary methods suggested by prior works, highlighting their strengths, limitations, and their implications in the context of this work.

## 3 PRELIMINARIES

### 3.1 NETWORK QUANTIZATION

In DNN quantization, values such as weight and activations (W/A) in the network are modified using a quantization function $Q(x) : \mathbb{R} \to \mathbb{R}$, so that $Q(x)$ can be represented using fewer bits. The benefits of quantization are two-fold: First, by reducing the amount of bits required to store weights and activations, quantization effectively compresses the model. This is especially important when working with deep-learning accelerators, since the memory requirement of the device is a common limiting factor for using large models. Second, quantization enables the utilization of cheaper General-Matrix-Multiplication (GeMM) engines in hardware. Therefore, the computation of quantized neural networks will require less energy and time, provided the hardware supports multiplication with the quantized format. There are many types of quantization methods (Guo, 2018), which depend on the specific goals, with different works focusing on different phases of training, different sources of information, or different formats.

In this work, we are interested in Post-Training Quantization (PTQ) of the diffusion's denoiser model. Most prior works (Li et al., 2023; Yang et al., 2023; He et al., 2024) have aimed to do this using integer (fixed-point) quantization:

$$Q_B^{\text{INT}}(x, s) = \text{round}\left(\text{clip}\left(\frac{x}{s}, -2^{B-1}, 2^{B-1} - 1\right)\right) \times s \qquad (1)$$

where $s$ is a scaling factor, and $B$ is the number of bits used to represent the quantized values, $Q_B^{\text{INT}}(x, s)$. For every quantization function $Q$ and scalar $x$, we define the quantization error as $\delta(x) \equiv |x - Q(x)|$. To ensure that the quantization error $\delta_B^{\text{INT}}(x)$ is bounded for all probable values of $x$, it is important to correctly choose the scaling factor $s$. Consequently, integer quantization schemes often rely on model calibration, where training samples are used to gather internal model statistics, allowing optimal selection of scaling factors for every layer (or even channel) in the network. Once established, each scaling factor can often be absorbed into the preceding or following layers in the DNN, to hide the cost of the additional scalar multiplications.

As an alternative to integer quantization, some works (Kuzmin et al., 2022; Shen et al., 2023) use lower-bit floating-point formats instead. In floating point quantization, we represent each value using *sign*, *mantissa* and *exponent*, with a predefined partition of bits: 1 bit for sign ($x_{\text{sign}} = \frac{1}{2}(1 - \text{sign}(x))$), $E$ bits for exponent ($x_{\text{exp}} = \lfloor \log_2(|x|) \rfloor$) and $M$ bits for mantissa ($x_{\text{man}} = 2^{-M}\text{Round}\left(2^M\left(|x|2^{-x_{\text{exp}}} - 1\right)\right)$). The total number of bits then becomes $B = 1 + M + E$. When quantizing a higher precision floating point value to a lower precision format (with fewer bits), the quantization function can be described as:

$$Q_{M,E}^{\text{FP}}(x, b) \equiv (-1)^{x_{\text{sign}}} \begin{cases} 2^{2^E - b - 1}\left(2 - 2^{-M}\right) & x_{\text{exp}} > 2^E - b - 1 \\ Q_M^{\text{INT}}\left(x - 2^{-b-M}, 2^{1-b-M}\right) + 2^{-b-M} & x_{\text{exp}} \leq -b \\ 2^{x_{\text{exp}}}\left(Q_M^{\text{INT}}\left(x_{\text{man}} - \frac{1}{2}, 2^{-M}\right) + \frac{3}{2}\right) & \text{else} \end{cases} \qquad (2)$$

Here, $b$ is the exponent bias. For standard IEEE definitions, the exponent bias is set to $b = 2^{E-1}$. However, as seen in Kuzmin et al. (2022), non-default values of the exponent bias parameter can be used to scale low-precision floating point values to the representation range (with the factor of $2^{-b}$), in a similar manner to how the scaling factor ($s$) was used for integer quantization. We denote the floating point quantization format as "M$x$E$y$", where the $x$ specifies the value of $M$ and the $y$ specifies the value of $E$.

In general, floating point quantization is considered to be more expensive (on hardware) than integer quantization (van Baalen et al., 2023) when using the same amount of bits. It also has smaller absolute

quantization errors for values in the representation range. However, floating point representation also has an extended range of representation, making it more robust to changes in the network's statistics. Furthermore, floating point quantization has a lower *relative* quantization error ($|\delta(x)|/|x|$) for the small values (in range), which may result in more accurate outputs in some networks.

**Stochastic Rounding** One particular quantization method that is important to this work is *stochastic rounding* (Gupta et al., 2015). For integers, stochastic rounding (SR) replaces the round operation in Eq. (1) with the stochastic function:

$$\text{SR}(x) = \lfloor x + n \rfloor, \quad n \sim \text{Uniform}(0, 1). \tag{3}$$

Stochastic rounding can be used for integer or floating-point quantization (Wang et al., 2018)). For floating point, we simply use SR for integer quantization ($Q_M^{\text{INT}}$) when implementing Eq. (2). As shown in Chmiel et al. (2022), when applied on a linear layer, SR increases the mean-square-error (MSE) when compared with round-to-nearest (R2N) rounding. Importantly, since $\mathbb{E}[\text{SR}(x)|x] = x$, SR also reduces the quantization bias. Therefore, stochastic rounding was used in DNN training to reduce bias in gradients, which helps optimization (e.g., Chen et al. (2020); Chmiel et al. (2022)). As a direct consequence, stochastic rounding has been implemented in the hardware of modern accelerators (Alben et al., 2020; Mikaitis, 2021), albeit it is not used by default in training and was never recommended for inference, to the best of our knowledge. Instead, modern quantization methods often aim for direct minimization of the quantization error (usually, by relying on training data), as is the case with methods such as AdaRound (Nagel et al., 2020), AdaQuant (Hubara et al., 2021) and BREC-Q (Li et al., 2021).

## 3.2 LATENT DIFFUSION MODELS

Diffusion models (Sohl-Dickstein et al., 2015), and latent diffusion models in particular (Rombach et al., 2022), are currently considered to be the best tools for the task of image generation. The majority of the experiments in this work will be performed over the Stable Diffusion XL (SDXL) model (Podell et al., 2023) and Flux (Labs, 2024), which are two of the most popular open-sourced models for text-to-image generation.

By default, the image generation pipelines for either SDXL or Flux include the following stages: The text prompt is processed by a text encoder, and is represented by a condition tensor, $C \in \mathbb{R}_{\text{cond}}$. Then, starting from a random tensor in latent space ($X \in \mathbb{R}_{\text{latent}}$) and a noise schedule $\{\sigma_t\}_{t=0}^{T-1}, \{\gamma_t\}_{t=0}^{T-1}$, the base model (U-Net for SDXL or Transformer for Flux) is used to predict the image's 'noise', which is then used iteratively to update the latent image, as per Algorithm 1. In the case of SDXL, a refiner model replaces the base model after a fixed number of steps (80% by default), and proceeds with Algorithm 1 until the last timestep, where $\sigma = 0$. The output of this iterative process is then fed to a Variational Auto Encoder (VAE), which decodes the latent-space tensor into a human-compatible image.

---

**Algorithm 1** Diffusion with Discrete Euler Scheduler ($\epsilon$-predicting), (Karras et al., 2022)

---

**Require:** $\{\sigma_t\}_{t=0}^{T-1}, \{\gamma_t\}_{t=0}^{T-1}, X \in \mathbb{R}_{\text{latent}}, C \in \mathbb{R}_{\text{cond}}$
  $\text{UNET}(x, \sigma, C) : \mathbb{R}_{\text{latent}} \times \mathbb{R}^+ \times \mathbb{R}_{\text{cond}} \to \mathbb{R}_{\text{latent}}$
  **for** $t$ in $0, 1..T-1$ **do**
    $\overrightarrow{D} \leftarrow f(X, \sigma_t) \equiv \text{DNN}\left(\frac{1}{\sqrt{\sigma_t^2+1}}X, \sigma_t, C\right)$
    $X \leftarrow X + \sqrt{\hat{\sigma}_t^2 - \sigma_t^2}\epsilon_t,$         with $\epsilon_t \sim \mathcal{N}(0, \mathbb{I}_{\text{latent}})$ and $\hat{\sigma}_t \equiv (\gamma_t + 1)\sigma_t$
    $X \leftarrow X - \Delta_t \overrightarrow{D}$         with $\Delta_t \equiv \hat{\sigma}_t - \sigma_{t+1}$
  **end for**
  **return** $X$

---

## 3.3 IMAGE SIMILARITY MEASUREMENTS

One of the main challenges when working with generative algorithms is the evaluation of the generated outputs. To avoid the expensive measurement of Mean-Opinion-Scores (MOS), most works utilize

numerical tools such as Frechet Inception Distance (FID) (Heusel et al., 2017), to assess the quality of the model output, despite the well-known limitations of these measurements for evaluating high-end models (it was shown to have a weak correlation with MOS with SDXL (Podell et al., 2023)). The mismatch between FID measurements and MOS may explain, to some degree, the limited impact that previous works had on the practical usage of quantized diffusion models thus far. To address this, we add one additional goal for our quantized diffusion model: That is, our goal in this work will be to produce the **same** image, as produced by the full-precision (FP) model. We believe that we have no better judge than the original model itself, when trying to evaluate the distribution of generated images of a quantized model.

To this extent, we will use the Peak Signal-to-Noise (**PSNR**) criteria as a tool to measure the similarity between the generated images, with either full precision or quantized model. For any image ($Y$) and it's reference ($X$), we calculate PSNR using:

$$\text{PSNR}(Y, X) = 20 \log_{10}\left(\frac{\max(Y)}{\text{RMSE}(Y, X)}\right). \tag{4}$$

PSNR values are measured in decibels (dB), with higher scores indicating a stronger match between the image and its reference ($\infty$ indicates an ideal match). For a more detailed analysis of the difference between images, we also include Structure Similarity Index Measurements (SSIM) in Appendix E.

## 4    ALLEVIATING QUANTIZATION BIAS

Our main interest in this work will be the quantization of the denoiser neural networks (U-Net or Transformers). The quantization will be applied to the values of all activations and weights, excluding the first and last layers. In accordance with previous works (Shen et al., 2023), we will use varying 8-bit Floating-Point formats, in combination with an optional flex-bias technique. The flex-bias we used allows cheap scaling for tensors, which is especially helpful when dealing with range-limited data types. Flex bias, as well as our choice of different quantization formats, is discussed in detail in Appendices B and C.

In Fig. 1a, we evaluate the PSNR scores of a naively quantized SDXL (see Appendix F for implementation details). Our results indicate that in terms of PSNR, the M4E3 format (4 Mantissa, 3 Exponent) is the best at staying close to the Full-Precision model, on the strict condition that values are scaled with flex-bias to compensate for the smaller range. We also note that, in all cases, the PSNR score tends to decrease as the number of steps increases. This is expected, since the more steps we take, the more quantization noise we add to the process, and the further we get from the process we had with full precision.

### 4.1    QUANTIZATION MSE AND BIAS IN DIFFUSION MODELS

The effect of quantization noise over a neural network (denoted as $f$), is often (He et al., 2024) modeled as an addition of a random variable ($z_t$) to the network's output:

$$f_Q(X_t, \sigma_t) = f_{\text{FP32}}(X_t, \sigma_t) + z_t. \tag{5}$$

It is important to point out that when using standard quantization with round-to-nearest (R2N) rounding, the quantization noise for a given input ($X$) is deterministic (it can only be modeled as stochastic by regarding the input as random). Therefore, the bias is not properly defined and while it is tempting to model $z_t$ as zero mean white noise, it is usually inaccurate. For example, He et al. (2024) has shown that $z_t$ is strongly correlated with the neural network's output.

In contrast, the quantization bias is defined and measurable for stochastic rounding (Eq. (3)). When applied over a single scalar within the quantization range, stochastic rounding guarantees unbiased quantization. Due to the effect of non-linearities, this statement is no longer precise when quantizing an entire deep neural network. From Eq. (5), we can calculate both RMSE ($r_t$) and average bias ($\bar{\mu}_t$) of SR-quantized U-Net using

$$
\begin{aligned}
r_t &\equiv \sqrt{\mathbb{E}\,\|z_t\|^2} = \sqrt{\mathbb{E}\left[\|f_{\text{Q-SR}}(X_t, \sigma_t) - f_{\text{FP32}}(X_t, \sigma_t)\|^2\right]}, \\
\bar{\mu}_t &\equiv \|\mathbb{E}[z_t]\| = \|\mathbb{E}[f_{\text{Q-SR}}(X_t, \sigma_t)] - f_{\text{FP32}}(X_t, \sigma_t)\|
\end{aligned}
\tag{6}
$$

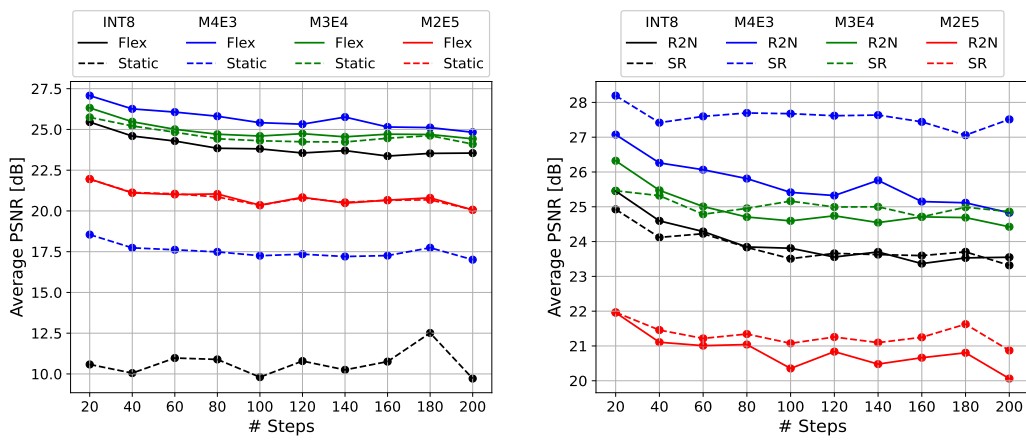

(a) Naive SDXL quantization using different formats      (b) SR vs R2N Rounding for Activations

Figure 1: Image Similarity scores with different formats. For each step count and format, we generated 100 images using the stable-diffusion-prompts dataset (Gus, 2023), comparing each image with the image generated by a full-precision model, with the same initial conditions. In all experiments, weights were quantized to the same format as the activations, using R2N rounding. We reported the average PSNR score, for each number of steps and quantization format. On the left panel, we compared static and flex-bias, showing that our implementation of online flex-bias is essential for the FP8-M4E3 format, and moderately helpful at improving PSNR with M3E4. On the right panel, we compared stochastic Rounding (SR) and Round-to-Nearest (R2N) rounding methods, for activation-quantization. When using inference with less than 200 steps, only networks with M4E3 data types consistently benefit from using SR. For SSIM evaluation of the same images, see Fig. 5.

where $\|\cdot\|$ denotes the L2 norm over the spatial dimensions, with implied normalization by the number of elements. For R2N, $r_t = \bar{\mu}_t$. In Fig. 2, we compare the values of $\bar{\mu}_t$ and $r_t$ for SR and R2N rounding of the activations. Our measurements of the network output's quantization error and bias are consistent with the known effect of SR over a single, linear layer: SR increases the overall quantization error $r_t$ when compared with R2N, but decreases the bias $\bar{\mu}_t$, to a substantial degree. The small bias measured for SR may seem surprising, given that many of the layers in the U-Net are non-linear. However, for a sufficiently small quantization noise, these layers become approximately linear with respect to the SR noise.

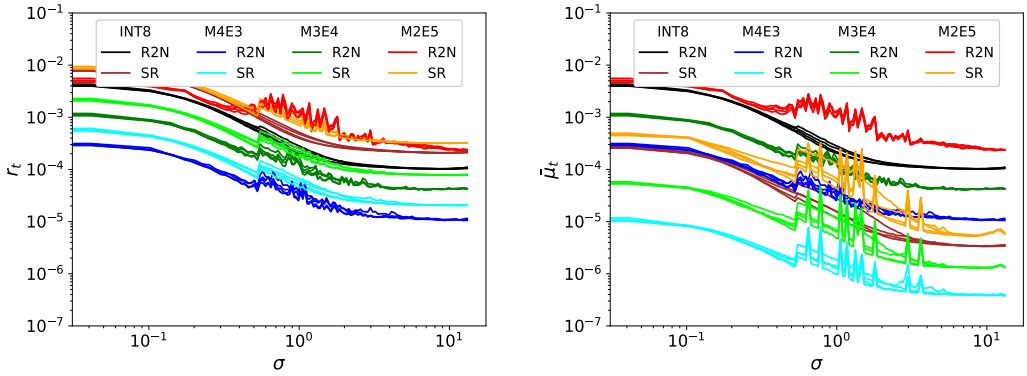

Figure 2: Quantization RMSE and bias in quantized neural networks (activations only) with R2N and SR rounding. Lines of the same color indicate different prompts. We ran a total of 100 steps, repeating each step 64 times, resampling the noise for SR for each repetition. We tested 4 different prompts, and 4 seeds per prompt, for every quantization-level and rounding method. We report (Left) the quantization error $r_t$ figure and (Right) the bias $\bar{\mu}_t$. SR is shown to drastically decrease the bias in all runs, while also increasing the overall quantization error.

## 4.2 THE EFFECT OF DEBIASING ON THE DIFFUSION PROCESS

In most common applications of neural networks (excluding diffusion), a single forward iteration of the model is applied to each data segment. Therefore, there is no practical difference between the output's quantization error and its bias. The diffusion process, on the other hand, has the unique property of being *iterative*. In Appendix H, we use this property and a set of simplifying assumptions to derive an equation for the probability, in which an unbiased quantization will have a larger overall error, when compared with R2N quantization. Given $Y$, $Y_{\text{deterministic}}$ and $Y_{\text{unbiased}}$, the latent images generated by the full-precision model, deterministically quantized model, and unbiased quantization model respectively, we get:

$$P\left(\|Y - Y_{\text{unbiased}}\|_2 \geq \|Y - Y_{\text{deterministic}}\|_2\right) \leq K \exp\left(\frac{-\bar{c}}{2\alpha}T\right), \tag{7}$$

with $T$ being the number of denoiser inference steps, $\alpha$ portraying the increase in MSE ($r_t^2$) for the unbiased approach (As per the bias-variance trade-off), $\bar{c}$ being the average correlation between the denoiser quantization errors in different steps when using the biased approach (Empirically, $\bar{c} \simeq 0.15$ [Fig. 8]), and $K$ being some constant. In other words, the chances of the biased quantization method producing better (less distorted) images than a bias-free approach decrease exponentially as we increase the number of steps in the diffusion process. We note that the half-life of this exponential decay is $T_{\frac{1}{2}} \simeq 36$, based on the empirical values ($\alpha = 4, \bar{c} = 0.15$ for SDXL, see Appendix H). This implies that the exponential term is significant for relevant values of $T$.

To evaluate the benefit of bias-reduction in real settings, we measure the effect of stochastic rounding on a model (SDXL) with quantized weights and activations (of the same format) and show the results in Fig. 1b. For all floating-point formats, we observe that the improvement (in terms of PSNR) gained by using SR becomes greater as the number of diffusion steps rises, albeit in most cases the gap remains when using a realistic number of diffusion steps (#Steps < 200). Nevertheless, the one exception is for the M4E3 format — in this case, SR quantized models consistently outperform R2N models, achieving the highest average PSNR across all tested models.

## 4.3 STOCHASTIC WEIGHTS

While the bias of activation quantization was reduced substantially with SR, our fully quantized network is still expected to have a large bias as a result of the weight quantization. Most advanced approaches for weight quantization (Li et al., 2021; Hubara et al., 2021) rely on stable statistics and data, and are thus not relevant for our case. Meanwhile, using stochastic rounding as we did for activations is not practical for the weights: Unlike activations, weights are only quantized once (per model), and keeping the full-precision weight in device memory just to quantize it again in every step would defeat some of the purposes of weight quantization (e.g., reducing peak memory).

Therefore, our solution will take the middle ground, between proper stochastic rounding and hardware efficiency. Effectively, we aim to quantize the weights with hybrid R2N/SR quantization, $Q_{M,E}^{\text{FP-SR}}\left(Q_{M+p,E}^{\text{FP-R2N}}(w,b),b\right)$. To implement this, we first quantize the weights by rounding them toward zero, while also storing the $p$ following mantissa bits of every value. For the compatibility of this approach with existing software and potential hardware, we will separate the bits into two values (in two separate tensors): $w_q = Q_{M,E}^{\text{FP}}(w,b)$ will hold the FP8 values, while a second value $w_p$ will store the remaining mantissa bits in (unsigned) integer format. During forward propagation, rather than using the weight $w$, we will use $\hat{w}$, a stochastic rounded variant of $w$ that follows the formula:

$$\hat{w} = \begin{cases} \text{Integer-Increment}\,(w_q) & \text{, If } w_p > n \\ w_q & \text{, Else} \end{cases}, \quad n \sim \text{Uniform}\{0, 1, \ldots, 2^p - 1\}. \tag{8}$$

The integer increment operation is used to "push" the FP8 values to the next quantization state. As its name implies, the operation can be simply implemented by treating $w_q$ as an (unsigned) integer and incrementing it by 1, under the condition that we avoid the edge case of exponent-overflow (by setting $w_p$ to be zero, when $w_q$ has the maximal value). Overall, the relative cost of this method in terms of memory is a $\frac{p}{B}$ increase in the memory reserved for weights (compared to FP8), and an additional operation with complexity on par with a ReLU/Dropout operation for every weight tensor. We will refer to the new method as *stochastic weights*, or *WSR*.

The addition of stochastic weights is necessary and effective for the nullification of the correlations between quantization errors in different time steps, as we see in Fig. 8. Furthermore, the benefit gained from using stochastic weights can be seen in Fig. 3. In a clean setting with FP32 activations, using WSR has a similar effect to what we saw in the case of SR for activations (Fig. 2). When applied during a diffusion process, weight SR significantly improved PSNR. Even when using a single bit for removing bias, we observed better PSNR that typically improved further as we increased the number of inference steps.

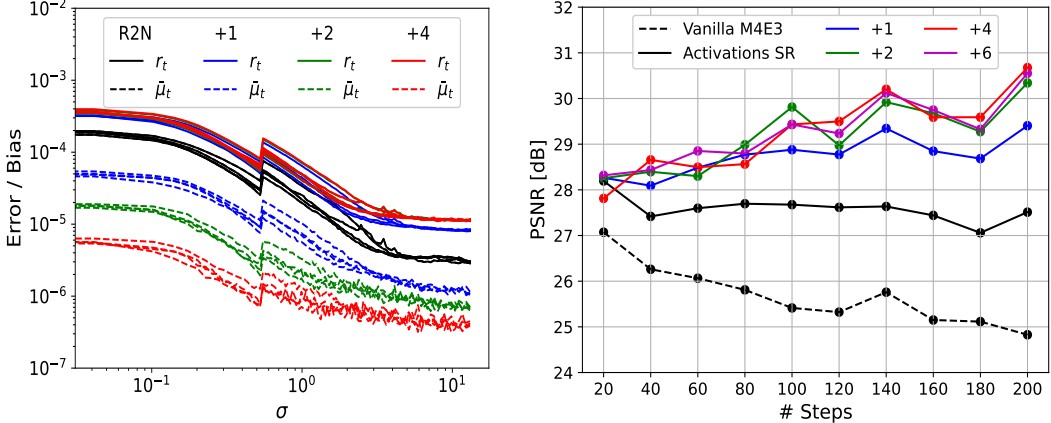

Figure 3: Quantized Diffusion model with "$+p$" stochastic weights. (Left) we measure the quantization RMSE ($r_t$) and bias ($\bar{\mu}_t$) in a setting where only the weights are quantized, with either R2N or stochastic weights (as per Eq. (8)). For this experiment, we used 4 different prompts, 2 seeds per prompt, and a total of 1024 repetitions per step, that were required to keep the results stable. (Right) we evaluate the average PSNR score when using M4E3 quantized models (W/A) with different levels of stochastic rounding, over 100 images generated from the stable diffusion prompts (Gus, 2023).

It is worth taking a moment to consider the implications of stochastic weights. After all, in terms of memory, we now operate with FP($8+p$) format (Floating Point with $8+p$ bits), and no longer using FP8 as intended. The reason we consider WSR to be helpful has more to do with the hardware restrictions: While FP($8+p$) $\times$ FP8 hardware GeMM kernels might have been preferable for our needs, it is not something we can expect to see implemented in hardware. This is because hardware kernels work better with balanced weights/activation bandwidth, and since there are no known floating-point standards that fit our use case (nor can we expect them in the near future). WSR can therefore be considered as a method to "squeeze" custom FP formats to hardware supporting FP8 datatype, gaining the benefit of cheaper computation operations and lower memory for activations (The activation memory, which still uses FP8 in our scheme, may exceed the memory used for storing weights by orders of magnitudes, as it scales with both batch size and image resolution).

## 4.4 ADJUSTED SCHEDULER

While most of the methods suggested in prior works are primarily aimed at integer quantization, the Variance Schedule Calibration (VSC) (He et al., 2024) may still be relevant for FP8 quantization. In VSC, the scheduler of the diffusion process is adjusted to take the constant injection of quantization noise into account, while injecting noise during DDPM/DDIM sampling. In our case, the Discrete Euler Scheduler (Algorithm 1) does not add noise by default ($\gamma_t = 0$), but we find that a careful adjustment of the scheduler can yield a consistent algorithm for a setting with quantization noise (i.e., consistent with the time-varying Langevin diffusion SDE, from Karras et al. (2022)). We use:

$$\gamma \simeq \frac{1}{2}\left(r_t^2 - \bar{\mu}_t^2\right)\left(1 - \frac{\sigma_{t+1}}{\sigma_t}\right)\left(3 + \frac{\sigma_{t+1}}{\sigma_t}\right). \tag{9}$$

Here, $\sigma_t$ are the scheduled noise quantities, that are provided by the scheduler. The full derivation is located in Appendix J. In our adjusted algorithm, $\gamma$ is not used to add artificial noise (The noise comes strictly from the quantization) but is still used when calculating the step size, $\Delta_t$. Measuring

Table 1: FID and PSNR measurements of 30K images generated by quantized SDXL models ($2.6B\times2$ parameters). Adding stochastic rounding to activations (SR) and weights (WSR) improves both FID scores and PSNR scores in all cases, with the possible exception of shorter, 20 steps experiments.

| Steps | 20 | | 50 (Default) | | 100 | | 200 | |
|---|---|---|---|---|---|---|---|---|
| | FID $\downarrow$ | PSNR $\uparrow$ | FID $\downarrow$ | PSNR $\uparrow$ | FID $\downarrow$ | PSNR $\uparrow$ | FID $\downarrow$ | PSNR $\uparrow$ |
| FP16 | 21.34 | $\infty$ | 19.45 | $\infty$ | 19.10 | $\infty$ | 18.82 | $\infty$ |
| FP8 | 21.58 | 25.86 | 19.60 | 24.49 | 19.11 | 23.84 | 18.95 | 23.60 |
| +SR | 21.56 | 26.12 | 19.27 | 25.40 | 18.88 | 25.02 | 18.72 | 24.99 |
| +WSR(p=4) | **21.10** | **26.29** | **18.88** | **26.26** | **18.51** | **26.92** | **18.37** | **27.74** |
| TRT FP16 | 20.67 | $\infty$ | 17.94 | $\infty$ | 17.46 | $\infty$ | 17.25 | $\infty$ |
| TRT INT8 | 22.48 | 22.49 | 48.94 | 18.87 | 555.4 | 13.88 | 570.12 | 12.75 |

$r_t$ and $\bar{\mu}_t$ does require calibration, albeit the quantization variance is independent of the full precision model, resulting in stable statistics that do not change for different prompts or initialization conditions. Nevertheless, we ultimately refrain from recommending the adjusted scheduler, as its effect on the scheduling process ends up being negligible ($\gamma \sim 10^{-6}$) for M4E3 quantization, and the calibration, however simple, may discourage users for using it.

## 5 EXPERIMENTS

So far, we have shown that a combination of M4E3 datatype, dynamic flex-bias, and stochastic rounding for weights/activations reduces the bias of quantization noise. Moreover, we showed this combination results in diffusion models that generate similar images to what we get with full precision models. In this section, we would like to evaluate the previously discussed methods on a larger scale, and show that put together, our methods constitute a valid alternative to the commonly used half-precision diffusion models.

As before, use the SDXL model with simulated quantization, switching from the base U-Net model to the refiner model after $80\%$ of the steps. We set all values and configuration to the default values for all hyperparameters, with the exception of the number of inference steps (50 by default), which we scan over. For evaluation, we generate $30K$ images with $1024 \times 1024$ resolution, using the $30K$ labels of the MS-COCO dataset (Lin et al., 2014), and a fixed seed (42). We scan over 20, 50, 100, and 200 inference steps ($T$), comparing quantized models with activation SR and weight SR ($p = 4$). The evaluation results, computed using the *text2image-benchmark* package Pavlov et al. (2023), are presented in Tab. 1. The full implementation details are included in Appendix F. For comparison, we also included the results for TensorRT (TRT), as discussed in Appendix A. In the case of TRT, we followed the default implementation, which did not contain a refiner network. Consequently (Podell et al., 2023), the baseline FID for TRT is lower.

In terms of the FID criteria, both stochastic rounding and stochastic weights resulted in a net benefit, even when the number of steps remained low ($N = 20$). Increasing the number of steps was also beneficial for FID in all cases. With the exception of $N = 20$, SR and WSR had lower FID than the baseline. This is surprising, as we had no expectation that a data-free method would surpass the trained model. We consider this to be the result of the FID criteria's limitations in the context of the SDXL model (Podell et al., 2023). The PSNR measurements were consistent with the results presented in the previous section, although scores with the MS-COCO prompts were generally lower than what we saw for stable-diffusion prompts. As we increase the number of steps, PSNR decreases when using R2N, remains steady when using stochastic rounding, and improves when also using stochastic weights. Due to the high similarity between all images, CLIP and Inception scores were less indicative, and are reported in Tab. 4, in the Appendix. Our subjective ranking, based on a few sampled images (for visual examples, see Appendix O and the attached supplementary materials), is consistent with the PSNR ranking.

In Appendix K, we included additional image-similarity experiments, with varying resolutions, guidance scales, and schedulers, showcasing the robustness of SR and WSR. In Appendix G, we implement our methods in the smaller (890M parameters) Stable-Diffusion 1.4 and show that they surpass Q-Diffusion (Li et al., 2023), even when tested in unfavorable settings.

Table 2: Similarity (PSNR ↑ / SSIM ↑) of images generated with quantized Flux models (12B parameters), when compared to images generated by the full-precision model. We used 50 diffusion steps to generate 100 images, based on stable-diffusion prompts and default workflow parameters.

| Format | Flex-Bias | Round-To-Nearest | | Stochastic-Rounding | | R2N + WSR (p=4) | | SR + WSR (p=4) | | Supported in Hardware? |
|---|---|---|---|---|---|---|---|---|---|---|
| M4E3 | | 16.58 | 0.64 | 16.60 | 0.65 | 21.93 | 0.78 | **22.21** | **0.79** | ✗ |
| M3E4 | | 23.40 | 0.82 | 23.24 | 0.82 | 24.60 | 0.85 | **25.00** | **0.85** | ✓ |
| M2E5 | | 20.94 | 0.76 | **22.20** | **0.79** | 21.72 | 0.78 | 21.81 | 0.78 | ✓ |
| M4E3 | ✓ | 25.64 | 0.86 | 26.26 | 0.87 | 26.99 | **0.88** | **27.01** | 0.88 | ✗ |
| M3E4 | ✓ | 24.32 | 0.83 | 24.57 | 0.84 | 24.72 | 0.84 | **24.78** | **0.84** | ✗ |
| M2E5 | ✓ | 20.87 | 0.76 | 22.25 | 0.79 | 21.55 | 0.78 | **22.34** | **0.80** | ✗ |

## 5.1 FLUX: QUANTIZING DIFFUSION TRANSFORMERS

The latest significant development for open-sourced text-to-image diffusion models was the release of Flux (Labs, 2024). Contrary to most diffusion models that rely on U-Nets, the denoising model in the Flux pipeline is a *diffusion Transformer*. While integer quantization of transformer models is exceptionally challenging (Xiao et al., 2023), Flux models that were naively quantized to FP8 datatypes have already been publicly distributed. Fortunately, the methods suggested in this work are agnostic to the specific architecture of the denoising neural network, allowing straightforward integration in any model. In Tab. 2, we test our methods over Flux, measuring the average PSNR score of 100 images generated with stable-diffusion-prompts.

In accordance with the results of the experiments over SDXL, using the M4E3 format with flex-bias achieved the best results, which were further improved by using Stochastic Rounding during the quantization of activations and weights in the model. By itself, the effect of SR for activations was less significant when implemented over the diffusion transformer, when compared with the more complicated, weight-SR, which resulted in a clear and consistent improvement. One immediate benefit of the Flux model is that naive M3E4 quantization is much more reliable, as opposed to its SDXL counterpart. As a result, our methods can be used to improve the output of the Flux model with M4E3 quantization implemented over existing, commercial hardware.

## 6 CONCLUSION

In this work, we have shown that it is possible to take advantage of the iterative nature of the diffusion process, in order to reduce quantization bias and enable faster, cheaper models. Our first novel recommendation — using SR during activation quantization, allows for almost cost-free improvement of the diffusion model's output. Our second recommendation, stochastic weights, can be used in addition to SR, trading small computation costs and small memory footprint for higher quality images. While our method works better for longer inference processes, we find that even shorter processes (20 steps) benefit from using it.

Image similarity evaluations, which we used to compare quantized and full-precision models, have proven themselves to be powerful tools for assessing the quality of post-training quantization. Despite their relatively small cost (only a few generated images), PSNR measurements were robust and highly predictive of FID scores that required $30K$ images to measure. We recommend using either PSNR or SSIM in all works that aim to make existing diffusion models more efficient. Research that is aimed at improving existing hardware, and therefore relies on slower software simulations, is likely to benefit greatly from using these tools.

Although some of the improvements suggested by this paper are tangible with existing hardware, the best results we have observed hinges on the availability of efficient, M4E3 GeMM computation in hardware. While none of the common FP8 formats have yet to receive an official IEEE standard definition, it appears that most hardware vendors prioritize the implementation of M3E4 and M2E5 formats. We strongly recommend prioritizing M4E3 implementation: as we have shown, the format not only performs better under some reasonable conditions but also opens the door for further improvements, which were not observed for the other FP8 formats.

**Reproducibility.** The paper fully discloses all the information needed to reproduce the experimental results. The full implementation details of the main results are detailed in Appendix F, and the full implementation details of the competitive methods are detailed in Appendix A. Statistical significance of the results is demonstrated in Fig. 9. The code used for generating the main figures and tables is included in the supplementary materials.

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

## A    LIMITATIONS OF EXISTING METHODS

In this section, we will empirically examine the image output of the most prominent alternative solution for 8-bit quantization of diffusion models, with a direct focus on its limitations. The significant shortcomings of the quantized models, as detailed in this section, may suggest why the majority of users are reluctant to take advantage of the existing quantization methods.

We identify two major limitations for the TensorRT implementation: First, we show that even when using the recommended settings, quantization of the SDXL model results in a decline in overall image quality, as measured by the FID criteria. Second, we show that the model does not maintain the high quality of the generated images when the number of steps is increased above 30.

In addition to the major limitations, we also explore the adaptability of the quantized models to different settings. We show that models that were calibrated using specific settings are less reliable when used with different parameters (The full precision model is expected to work with all parameters). We consider this to be a minor limitation, since the overall effect remained relatively small (The maximum increase in FID was 0.7).

### A.1    EXPERIMENT DETAILS

All the results shown in this section rely on the TensorRT implementation[1], and is officially associated with Q-diffusion (Li et al., 2023). By default, all parameters match the recommended settings, including 30 diffusion steps, Euler scheduler, and 'O3' quantization level (CNN + FC). Consequently, the default quantized U-Net does not include quantization of the QKV component during the forward propagation of the attention modules. We do not use refiner in any of the experiments in this section, as it is not currently included in the TensorRT implementation. The visual comparison of images in this section will based on 4 prompts:

- The Default Example prompt: "A photograph of an astronaut riding an horse."
- Stable Diffusion Prompt (without artist names): "detailed modern office space, future utopia, concept art, fine art, very detailed, very realistic"
- Coco Prompt: "baseball player is holding a bat in a baseball game."
- Custom Prompt: "A beautiful oasis in the middle of the endless void. Epic Realistic, (hdr:1.4), (intricate details, hyperdetailed:1.2), artstation, vignette, complex background"

### A.2    SCALING QUANTIZATION METHODS

In Q-diffusion (Li et al., 2023), the authors start by showing that 4-bit quantized models can maintain the quality of image generation in smaller unconditional models and models with a fixed amount of classes. For the larger stable diffusion models, some quality degradation was observed, even when applying 8-bit quantization. Our first goal in this section would be to investigate how the fine-tuned approaches work when the model is even larger, as in the case of the SDXL model. In Tab. 3, we start by using the default settings of the TensorRT library to generate different images. Even when fully quantizing the attention mechanisms, the resulting images are of high quality, but fall short of achieving the same quality as the full-precision model in terms of FID (Tab. 3).

### A.3    NUMBER OF DIFFUSION STEPS

The calibration process in TensorRT (as well as most of the data-dependent quantization approaches for diffusion models), is performed per-timestep. Consequently, a calibrated model for 30 steps is not intended to work with any other number of steps. This is a significant limitation, that prohibits the user from adjusting the number of steps on demand. Perhaps the most concerning issue is that the default number of diffusion steps in TensorRT is too small, when compared with the default hyperparameters in other libraries, like diffusers (von Platen et al., 2022). The default number of steps for stable-diffusion models, as well as the most recent Flux model, is 50. However, calibrating and running quantized TensorRT models with 50 steps reveals a very clear degradation in image quality,

---

[1]Example code, and full environment, can be found at `https://github.com/NVIDIA/TensorRT/tree/release/9.3/demo/Diffusion`

Table 3: Evaluation of TensorRT (INT8) with SDXL over different domains

| Quantization | Calibration | FID ↓ | CLIP ↑ | PSNR ↑ |
|---|---|---|---|---|
| FP16 | — | 19.34 | 0.32 | ∞ |
| INT8 'O2.5' (CNN+FC+QKV) | Default | 22.32 | 0.32 | 22.05 |
| INT8 'O3' (CNN+FC) | Default | 21.94 | 0.32 | 22.01 |
| | Scheduler → DDPM | 22.69 | 0.32 | 21.84 |
| | Scheduler → LCM | 22.36 | 0.32 | 21.91 |
| | Scheduler → LMSD | 22.33 | 0.32 | 22.13 |
| | Scheduler → PNDM | 22.18 | 0.32 | 21.73 |
| | Guidance Scale → 3.0 | 22.00 | 0.32 | 21.83 |
| | Guidance Scale → 9.0 | 22.55 | 0.32 | 21.94 |
| FP16 (50 steps) | — | 17.47 | 0.32 | ∞ |
| INT8 'O3' (50 steps) | Default | 48.95 | 0.31 | 18.85 |

as can be seen in Fig. 4. Simply put, the quantization method used in TensorRT is not serviceable when using the default SDXL hyperparameters. The results are significantly improved when using SDXL-Turbo, where the number of steps is extremely low, but SDXL-Turbo is not the standard mode of operation for diffusion models.

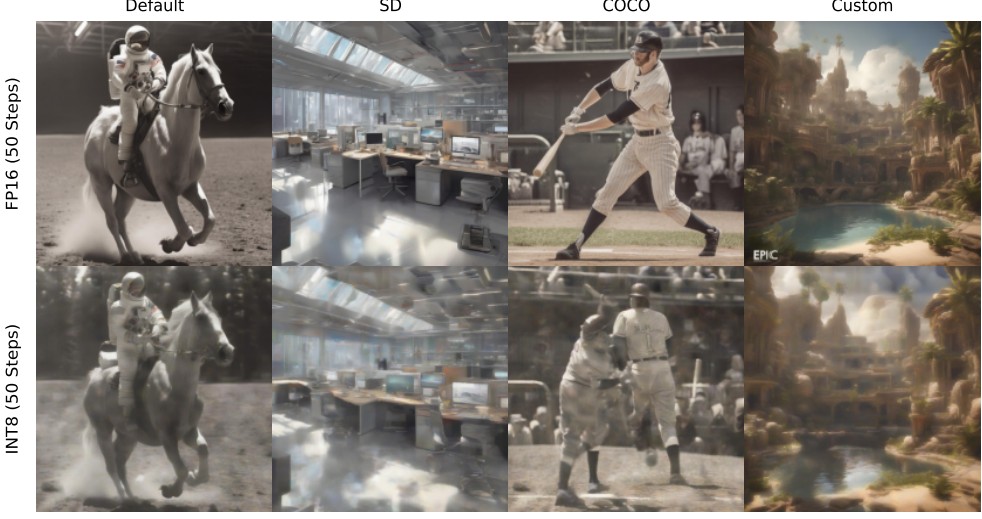

Figure 4: TensorRT SDXL with 50 Diffusion Steps.

## A.4 DOMAIN LIMITATION

All integer quantization methods require a calibration of the full-precision models, for the model values to be scaled, in preparation for operations with limited range, and larger numerical errors. Via calibration, the algorithms capture the statistics of the model, which allows better compression than what can be achieved by data-free models. However, reliance on calibration means that the internal statistics of the model must remain consistent. This is especially challenging for diffusion models, which are traditionally expected to work over different timestamps, different prompts, different schedulers (that include varying input normalization schemes), and different hyperparameters, without being re-calibrated.

Previous works (Li et al., 2023) have focused on the changes in statistics resulting from changing timestamps, albeit the rest of the diffusion model's parameters remained mostly unexplored in academic research. For example, several works have opted to use MS-COCO captions for calibration, while also using the same captions during the evaluation of the model. To check whether calibration

results in domain gaps, we opt to reproduce the results of the default TensorRT setting (Appendix A.2), but use different schedulers and key hyperparameters during the model **calibration**. Our results, as seen in Tab. 3, show that quantized diffusion models produce images of lower quality (Up to 0.75 increase in FID score) when the calibration hyperparameters do not fit the hyperparameters used in inference. This type of degradation will not occur in data-free settings that do not rely on calibration.

## B  CHOOSING THE QUANTIZATION FORMAT

The first important decision for any quantized neural network before any adjustment or correction, is the choice of quantization format. First, we need to choose an optimal partition between mantissa and exponent bits for the 8-bit floating point format. Second, we know from previous works (Shen et al., 2023) that scaling factors can be very helpful, especially in cases where the range is limited. However, scaling factors are traditionally calibrated using training samples, which are not available in data-free settings.

Instead, our scaling method will rely on online operations, that will be performed during forward propagation. While scalar scaling operations are not, in general, as expensive as GeMM operations (They have linear complexity, compared with the square complexity of GeMM), they may still induce significant computational cost. To mitigate this additional cost, we suggest the following, lightweight flex-bias (Kuzmin et al., 2022) scaling operation, which is performed once per quantized tensor:

$$b = 2^E - \max(x_{\exp}) - 1, \tag{10}$$

where the $\max$ operation is performed on all the exponent components ($x_{\exp}$) of the values in the quantized tensor. The main benefit of this approach, and what distinguishes it from implementations seen in previous works, is that we ensure that all scaling factors ($2^{-b}$) are power-of-two scalars, since $b$ is integer and shared per tensor. This will simplify any multiplication and addition operations between quantized tensors that have different scaling factors. The limited range of the values of $b$ can potentially be utilized in hardware to also reduce the cost of non-linearities. We note that the main benefit of using offline scaling factors is the ability to merge the factors in the network to avoid additional operations during forward propagation, but this method is not applicable when the factors have to be adjusted from step to step (as is the case for diffusion models (Li et al., 2023)). A more detailed analysis of the cost of this method with either software or hardware implementations is included in Appendix C.

Another key decision during model quantization is the choice of which elements we should quantize and which we should keep at full precision. Previous works have studied the effect of different components in the U-Net (Yang et al., 2023), and highlighted several components, with some (He et al., 2024) even going as far as changing the level of quantization per diffusion timestep. While these approaches can be very effective at reducing the computation time, one of the main goals in our work is the reduction in peak on-device memory, which necessitates maintaining as constant memory as possible (The peak memory will be determined exclusively by the steps using the larger data-types). As a result, we will follow the example of Shen et al. (2023): We quantize all Linear, convolutions, and Matmul operations, with the exception of the first/last convolutions in the U-Net, which are kept in full precision.

## C  FLEX-BIAS COMPLEXITY ANALYSIS

One of the main features that allows our conditional, text-to-image models to operate in data-free settings, is the implementation of lightweight flex bias. In this section, we break down the cost of each operation as a result of the use of the flex-bias. For ideal efficiency, each computation element will be implemented in hardware, but for practical reasons, we also estimate the cost of the software implementation of each component, using common, available hardware. We note that most of the analysis is not unique to our method of scaling, but it may still be helpful to be reminded of the effects scaling factors have over the network.

For all matrix operations, we take $D$ to be the number of elements in the tensor, $E$ to be the number of exponent bits, and $M$ to be the number of mantissa bits, with $B = M + E + 1$.

For our software analysis, we will treat all operations excluding reads (fetch) and multiplication (mul) as having the same cost as addition, $\text{ADD}_B(1)$. For our hardware analysis, we assume that operations

are integrated within existing hardware components, meaning that the cost of fetching the data is considered to be zero, and it is possible to operate on specific bits. All estimates in this analysis will be rough, as the goal is to prove the feasibility of our approach and highlight implementation priorities, and not to suggest a specific design. Despite using similar terminology, we think of the software cost in terms of *operations*, while the hardware cost is considered in terms of *gates*.

As a point of reference, remember that the majority of computation in every neural network occurs during general matrix multiplications, and has a complexity of $\mathrm{MUL}_{M,E}\left(D^2\right)$. An estimation of the cost of $\mathrm{MUL}_{M,E}$ for GeMM operations can be found in van Baalen et al. (2023)– as a general rule the values for all 8-bit formats are similar, but it is cheaper to have more bits allocated to the exponent. In any case, the cost of multiplication is greater than the cost of integer addition, and $D^2 \gg D$.

### C.1 ONLINE EXPONENT-BIAS DISCOVERY

As we saw in Eq. (10), the formula we used for finding the flex bias during forward operation is $b = 2^{E-1} - \max\left(x_{\exp}\right)$.

In a software implementation, it is not possible to distinguish the sign, exponent, and mantissa. The straightforward approach would be to find the maximal value in the tensor by comparing all absolute values and extract the mantissa value from the scalar we get. The approximate cost would be therefore be $\mathrm{ADD}_B\left(D\right)$.

With dedicated hardware, we have the option to only look at the bias exponent when calculating the exponent bias. Consequently, the cost in this case would be estimated to be around $\mathrm{ADD}_E\left(D\right)$.

Of course, it is also possible to forgo the online gathering of statistics, calibrate the model offline, and use values from memory. With the cost of increased complexity of this approach from the user view (Forces calibration with data that may not be available) and the risk of bad results due to the difficulty of calibration (Different prompts may result in different statistics, and it is always possible that out-of-distribution prompts may affect the results), using an offline method will reduce the cost of this operation to a single scalar fetch operation. The cost of the fetch operation will depend on the accelerator architecture, but may not be trivial, since the scaling factor is expected to change in every timestamp for every tensor (Li et al., 2023), and existing accelerators don't necessarily have the capacity to fetch a single scalar with low latency.

### C.2 QUANTIZATION

The value of flex-bias $b$ works as a scaling factor for the quantized value.

Of course, there is no practical reason to work with quantized neural networks, without implementation of the quantization in hardware. During software simulation, we multiply each value in $2^b$ before simulating quantization using the qtorch (Zhang et al., 2019) library and multiply the number by $2^{-b}$ afterward.

For hardware implementation, $b$ is used to clip the values of the mantissa, to the range $[-b, 2^E - b - 1]$. This would mean first shifting the value of the unquantized exponent ($8 - bit$ with full precision) by $b$ ($\mathrm{ADD}_8\left(D\right)$), and performing the quantization with the updated values.

### C.3 TENSOR MULTIPLICATION

In either software or hardware implementation, multiplying two tensors with a power-of-two scale factor has a negligible cost.

### C.4 TENSOR ADDITION

Addition operations are often overlooked in the context of quantization, despite being very common in U-Nets, and being generally more problematic when performed in combination with scaling factors.

When adding two tensors in either software or hardware, we must first rescale the factors of one of the tensors to match the factors of the other, before performing the addition. With a power-of-two scaler, the scaling itself does not require multiplication and will have roughly the same cost as addition, $\mathrm{ADD}_B\left(D\right)$. The $B$ factor is relevant for hardware as well, because despite only operating over the

exponent, we may still need to set the mantissa in the cases where the operation over the exponent cause overflow ($x_{\exp} > 2^E - b - 1$) or underflow ($x_{\exp} < -b$). In hardware, it is also possible to leverage the lower range of the tensors by having a dedicated addition operation for tensors with different scales.

### C.5 NON-LINEARITY

For activation functions implemented in software, the input values must be in full/half-precision format and will have the same complexity of full-precision operation. In this case, there is no benefit to quantizing the number before the activation. If we do so for simulation purposes, we must rescale the value before the activation by multiplying it by $2^{-b}$ ($\text{ADD}_{16}(D)$), perform the activation, and quantize the output of the activation operation once again.

For hardware, the implementation is straightforward for homogenous functions like ReLU ($\text{ADD}_B(D)$). Polynomial functions and lookup table implementation are expected to benefit as well, but their actual gain will depend on the activation function itself.

### C.6 NORMALIZATION

Normalization is straightforward: we can simply terminate the scaling factor and proceed to the normalization as is.

## D BRIEF SURVEY OF METHODS PROPOSED BY PRIOR WORKS

In this section, we detail all methods suggested by prior works. For all components, we provide a brief summary of costs and benefits. For each component not included in this work, we explain our reason for not including it.

### D.1 INTEGER QUANTIZATION

Integer quantization was used in most prior works (Li et al., 2023; He et al., 2024; Yang et al., 2023). The immediate benefit of integer quantization is that it is more efficient from a hardware perspective, and is already included by default in most hardware accelerators. Integer quantization also exhibits smaller errors while the values are in the range of representation. The downside of integer quantization is that the range itself is limited, and once values are out of range, the error rises drastically. This tends to lead to lower accuracy (Fig. 1a and Fig. 1b). Consequently, we consider integer quantization to be less robust and preferred using FP8 formats that have an extended range. Due to the larger range, FP8 quantization is less prone to large errors caused by out-of-distribution values. Our choice to use FP8 was also influenced by the better, zero-shot accuracies we observed for FP8 formats.

### D.2 TIME STEP-AWARE CALIBRATION (LI ET AL., 2023)

Time step-aware calibration directly addresses the problem of changing statistics as a result of changes in $t$, which was observed in diffusion models. As the name implies, in time-step aware calibration we re-calibrate the quantization (e.g. finding $s$ from Eq. (1)) for all layers/components, for different values of $t$. In Li et al. (2023), the authors used a small set of samples to re-calibrate the model. It should be noted that re-calibration has a hidden cost on performance: we can no longer "merge" the scaling factors into the network in any natural way, meaning that you must either reload weight values (not recommended) or perform the scaling operation explicitly, paying the cost of extra, scalar-vector multiplications.

In our work, we insisted on a data-free approach and were generally careful of relying on statistics that depend on prompts (It's not clear how one can sample the prompt space to begin with). In general, FP8 quantized networks are more robust to changes in scaling, enabling the use of our power-of-two, flex bias implementation instead of time-step aware calibration. Our flex-bias implementation is also time-step-aware (it's performed online), and has the additional cost of finding the min/max of each tensor, but it is more software/hardware friendly in other aspects (e.g. rescaling tensors during addition is cheaper).

### D.3 SHORTCUT-SPLITTING QUANTIZATION (LI ET AL., 2023)

In shortcut splitting quantization, specific tensors in the U-Net are broken down into two tensors, each of them using a different scaling factor (Scaling factors are usually selected per output channel, for easy multiplication). While the authors give a convincing argument for why the added scaling factors only have a small effect on the following matrix multiplication operation, we note that in the case of SDXL model, the values of all candidates for shortcut-splitting quantization were also passed along through residual connections. This adds additional difficulties, since two scaling factors will now be needed for all tensors during the upsampling process–otherwise, we will lose the information we tried to preserve after a single addition operation.

As a result, we decided not to use shortcut-splitting quantization, and rely on the extended range of the M4E3 format instead.

### D.4 CORRELATED NOISE CORRECTION (CNC) (HE ET AL., 2024)

He et al. (2024) makes the observation that the quantization error in $\epsilon$-predicting quantized U-Net is highly correlated with the output of the full precision U-Net, possibly as a result of the multiple normalization layers in the network, that entangle the error with the clean output. This makes the attempt to remedy the quantization noise and integrate it into the diffusion process more difficult since we can not regard it as white noise (which we need for DDPM/DDIM). Correlated Noise Correction is designed with this in mind. First, we collect statistics to estimate the correlation in every timestep, and then, apply it to better "reconstruct" the clean output from the quantized network's output.

Despite CNC being relevant in combination with VSC, and despite observing the same correlation as reported by He et al. (2024) in our setting, we forgo implementing CNC when designing our adjusted scheduler (Sec. 4.4), for three reasons. First, since the network output is much less biased with SR quantization, this decreases these correlations, as we explain in Appendix L. Second, based on the values of correlation we measured, CNC would mean an amplification of the quantization noise, which can result in a diffusion process that is too far away from the full-precision diffusion process (and thus would go against our SSIM target). Third, we observed that the correlation statistics were very unstable, and were affected greatly by changing seeds and prompts. Therefore, we preferred not to rely on these statistics and so did not use this method.

### D.5 VARIANCE SCHEDULE CALIBRATION (VSC) (HE ET AL., 2024)

In Variance Schedule Calibration, the scheduler is adjusted to address the addition of quantization noise to the process (DDPM). This adjustment eventually translated to changes in the noise schedule ($\sigma_t$). In Sec. 4.4, we suggest different adjustments to the Discrete Euler scheduler. While the surface goal is similar in both cases (to address quantization noise), the difference in the schedulers is substantial. As a result, our adjusted scheduler maintains the original noise-schedule but has a modified step size ($\Delta_t$).

In the end, we decided not to recommend an adjusted scheduler. The adjusted scheduler was reliant on statistics, which while stable, still went against our data-free approach. Furthermore, the absolute effect on step size was negligible and we could not show a significant advantage (with the SSIM metric) when using it.

### D.6 BIAS CORRECTION (BC) (HE ET AL., 2024)

Despite its simplicity, Bias Correction was shown to be one of the more effective methods (Table 1,He et al. (2024)) for network quantization. In bias correction, we first calibrate the network using a subset of training samples, measuring the error in the network's output for every time step (And in the case of the conditional model, for every class). The statistics are then used during inference to "correct" the output of the network in each step.

Of course, this makes Bias-Correction unfeasible in data-free settings, especially for text-to-image models, where the number of classes is effectively infinite.

### D.7 Step-aware Mixed Precision (He et al., 2024)

Changing the quantization precision for different values of $t$ during inference is no doubt effective at reducing the quantization noise in the vital steps in which the image's final form is shaped. However, changing the precision during inference also means that the peak memory is determined solely by the memory in the steps where the highest precision was used. We avoid using step-aware mixed precision for that reason.

### D.8 Block Sensitivity Identification and Calibration (Yang et al., 2023)

To mitigate the wide, dynamic range of activations, Yang et al. (2023) suggested studying the quantization noise (via SQNR), and enhancing precision on demand on bottleneck components. In our case, we also used hybrid quantization, since the first and last convolution layers in our setting are kept in half precision. However, changing the precision of internal operations may increase the peak memory (the component with higher precision may become the bottleneck). In a network with as many residual connections as U-Net, bottlenecks are abundant, so special care is required for the selective quantization of layers. Lastly, SQNR relies on training samples, and so is not data-free. For these reasons, we avoid using this method in our work.

## E    Similarity Measurements via SSIM and FID-32

### E.1    SSIM

Out of the many different methods to measure similarity between images, we chose to use PSNR as the key measurement. PSNR is commonly used to measure differences between outputs, as it benefits from being simple (Both in terms of being easy to understand, and being easy to compute). Nevertheless, it is also important to acknowledge that, by virtue of PSNR's simplicity, it may overlook subtle details in images that are not captured through a basic pixel-to-pixel comparison. To address this, this section will include additional results for image similarity, that are based on the Structure-Similarity-Index Measure (SSIM) (Wang et al., 2004) score.

SSIM evaluates the perceived change in the structural information between two images. Its scores range from $-1$ to $1$, with $1$ being the ideal score, indicating a complete match between the image and its reference. For baseline numbers, we observe that the FP16 variant of SDXL generates images with SSIM of $0.973 \pm 0.002$ when compared with the FP32 model (For BF16 quantization, the SSIM are in the range $0.948 \pm 0.005$). In the context of this work, we note that even with an SSIM value of $0.8$, some effort is required to distinguish the image from its reference, while an SSIM of $0.6$ indicates that the images are substantially different. (See Appendix O for illustrations).

### E.2    FID to FP32

One additional, novel way to measure similarity is FID-FP32, which was proposed by Tang et al. (2025) in the context of the quantization of diffusion models. Like FID, FID-FP32 measure the Fidelity Inception Distance, but uses the outputs of the full-precision model as the baseline the distance measurement, rather than using the ground-truth (such as the images of the MS-COCO dataset). Consequently, FID-FP32 measures similarity between the images of the quantized and non-quantized models, with emphasis on the features of the Inception neural network.

In figure Fig. 5, we present the PSNR and FID-32 measurements corresponding to the experiments presented in Figs. 1a, 1b and 3 and Tab. 1. All similarity metrics (SSIM, PSNR and FID-32) are in strong agreement regarding the similarity in all experiments. The main exception is that with PSNR evaluation, we can see a small benefit for using SR with all floating-point formats when the number of steps exceeds $80$. The small gap between SR and R2N rounding was not observed for M2E5 and M3E4 when we used SSIM as a metric, as we saw in Fig. 1b.

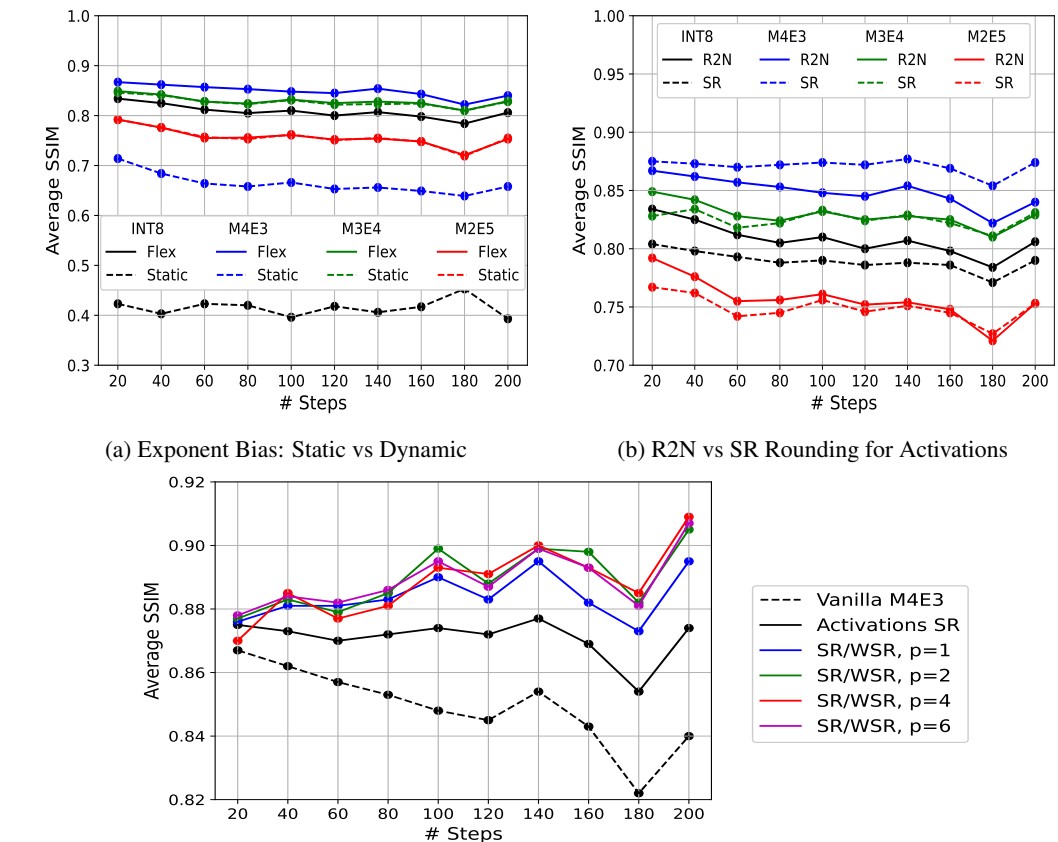

(a) Exponent Bias: Static vs Dynamic

(b) R2N vs SR Rounding for Activations

(c) R2N vs SR Rounding for Weights and Activations

| Steps | 20 | | 50 (Default) | | 100 | | 200 | |
|---|---|---|---|---|---|---|---|---|
| | FID-32 ↓ | SSIM ↑ | FID-32 ↓ | SSIM ↑ | FID-32 ↓ | SSIM ↑ | FID-32 ↓ | SSIM ↑ |
| FP16 | 0.28 | 1 | 0.18 | 1 | 0.17 | 1 | 0.17 | 1 |
| FP8 | **1.14** | 0.84 | 1.06 | 0.82 | 1.05 | 0.82 | 1.05 | 0.82 |
| +SR | 1.19 | 0.84 | 1.02 | 0.83 | 0.99 | 0.83 | 0.97 | 0.83 |
| +WSR(p=4) | 1.16 | 0.84 | **0.94** | **0.85** | **0.82** | **0.86** | **0.72** | **0.87** |
| TRT* FP16 | 0.0 | 1 | 0.0 | 1 | 0.0 | 1 | 0.0 | 1 |
| TRT* INT8 | 2.16 | 0.75 | 39.61 | 0.66 | 465.96 | 0.43 | 614.48 | 0.54 |

\* For TensorRT, we used FP16 as the baseline for the FID measurement.

(d) FID and PSNR results for the MS-COCO prompts experiment ($30K$ images)

Figure 5: Image similarity scores for stable-diffusion prompts, measured with SSIM. Figures (a),(b),(c),(d) correspond with the experiments conducted in Figs. 1a, 1b and 3 and Tab. 1. In all cases, the SSIM values are strongly correlated with the PSNR values.

## F EXPERIMENT DETAILS

All the experiments in this paper (excluding the Robustness check in Appendix K) rely on the default working flow (*diffusers* library (von Platen et al., 2022), version $0.25.1$) of the stable-diffusion XL model, with added refiner. All images were generated with a guidance scale of 5.0, in $1024 \times 1024$ resolution. For the base pipeline, we used the Huggingface model *stabilityai/stable-diffusion-xl-base-1.0*, which includes a Text-Encoder, Denoiser U-Net, Discrete Euler Scheduler ($\forall t, \gamma_t = 0$) and Variational Auto Encoder. For the refiner, we *stabilityai/stable-diffusion-xl-refiner-1.0*. For image generation, the base model U-net operated on the first $80\%$ steps, while the refiner worked on the last $20\%$. The positive prompt was used for both base and refiner models, and the negative prompt, when provided, was used for the base model only.

Every image was generated on a single NVIDIA GPU, either RTX A6000 or A100. To ensure consistency between experiments, we used a constant batch size of 1 in all experiments and initiated a generator with a pre-determined seed before every run.

In all quantized runs, we used the half-precision variant of Huggingface pre-trained models, which has a $2\times$ faster runtime and lesser memory footprint, enabling the simulation-heavy experiments (like stochastic weights). The reference image was generated with a full-precision model for the experiments using stable-diffusion prompts, and half-precision for the experiments using MS-COCO prompts. We note that using FP16 images for reference did not have a significant effect on the results, despite the moderate dissimilarity that was observed when comparing the FP32 images with FP16 images (SSIM of $\sim 0.97$). For INT8, our flex-bias implementation was not, by itself, sufficient for proper image generation. To ensure a fair comparison, we enhanced the INT8 model by adding scaling parameters per channel.

Model quantization was done via simulation. We used *qtorch* (Zhang et al., 2019) library to quantized datatypes to either INT8 or FP8 datatype, with either stochastic or round-to-nearest datatype. The quantization was performed prior to all GeMM operations: all linear operations and convolution operations in the network, or the Matmul operation in all attention heads. The only exceptions were the first and final convolution modules, *conv-in* and *conv-out*. The output of GeMM operations, as well as the input of other operations, was not explicitly quantized: This was done to avoid simulation overhead, since the quantization of elements such as activations isn't considered to be a problem, and since the memory usage only reaches its peak during GeMM operations, in normal circumstances. Flex bias was implemented by multiplying the value by the scaling factor prior to quantization, and then quantizing it. Of course, this method is only possible in simulation when the actual value is still stored in FP16 format, and not possible in hardware-compatible implementation, in which case the flex-bias needs to be tracked as well (see: Appendix C).

For the experiments gathering statistics from the diffusion process, we encapsulated the U-Net in the RepeatModule. For every step, the RepeatModule first disabled activation quantization, restored full-precision weights from memory and ran the model in full precision. Then, the activation quantization was re-enabled, weights were quantized again, and we ran the forward mode multiple times (64 for activation quantized networks, 1024 for weight quantized networks). $\bar{\mu}_t$ and $r_t$ statistics were measured from the results of the FP32 model and quantized model. For the next step, we always used the results of the FP32 model.

For stochastic weights, quantized weights using R2N to an intermediate data size, with $M + p$ mantissa bits and $E$ exponent bits, and stored it on device. Then, for every step, we used the weights with values computed by quantizing the intermediate data size yet again, using stochastic rounding this time. This scheme should have the same effect as following Eq. (8).

For the prompts, we used either stable diffusion prompts from the *Gustavosta/Stable-Diffusion-Prompts* huggingface dataset (Gus, 2023), or MS-COCO labels (Lin et al., 2014). For stable-diffusion prompts, we use the 100 first prompts in the dataset, and used incrementing seeds for the random generators fed to each image, starting from 0. For MS-COCO, we used a fixed seed (42), to simplify the fragmentation of the large generation task to different compute nodes.

FID and CLIP evaluations were performed using the tools provided by Pavlov et al. (2023). After generating images with the 30K labels of the MS-COCO dataset, FID is computed based on the images in the dataset, and CLIP is computed in relation to the labels. SSIM score was evaluated either using *pytorch-mssim* library (on device) or using *skimage* library (on CPU). The inception score was evaluated using torchvision's *inception-v3* model, with upscaling of $299 \times 299$ (bilinear), with a batch size of 32.

For the examples images (Fig. 12 and Fig. 13), we searched for still-life/ landscape images, with clear visual distinctiveness. The prompts selected from stable-diffusion prompts and MS-COCO labels were #89 and #126065 respectively.

## G    STABLE DIFFUSION 1.4 AND DIRECT COMPARISON WITH Q-DIFFUSION

In the following section, we extend our methods to support additional Stable Diffusion models (SD1.4), to showcase that our methods generalize to other settings. Unlike SDXL, SD1.4 were shown

to be efficiently quantized to 8-bit datatypes (i.e., quantized without significant reduction in FID), and can therefore be seen as an easier benchmark for diffusion model quantization. In addition to the smaller scale, one key difference between the SD1.4 pipeline and the SDXL pipeline is that the SD1.4 does not use SDE based scheduler as default. Consequently, we consider the experiments over SD1.4 to be a "worse-case" scenario for our methods. However, we will show that even when using default hyperparameters that were tuned for Q-diffusion, our methods results in better images with fewer mistakes and more similarity to the original, full precision images.

In prior sections, we relied on the Tensor-RT implementation as baseline, since no other official implementation of Q-Diffusion over SDXL was available. In the case of SD1.4, we can work with the official Q-diffusion implementation directly. We use the recommended setting for Q-diffusion, with 8-bit weights and activations, excluding the softmax activations in the attention that still relies on 16-bit values. All experiment perform 50 steps of diffusion using the PLMS scheduler. Our results– in the form of the average PSNR over 100 images and a visual illustration of the first 7 generated prompts (according to the dataset's default order), are shown in Fig. 6

## H   Convergence of the Diffusion Process

Given a model $f(X, \sigma)$ (with implicit conditioning), a timesteps scheduler $\{\sigma_t\}_{t=0}^{T-1}$, an initial state $X_0$, and quantization noise $\{z_t\}_{t=0}^{T-1}$, the diffusion process and the quantized diffusion process can be described via the states $\{X_t\}_{t=0}^{T-1}$, with

$$
\begin{array}{rcl}
X_{t+1} & = & X_t + \Delta_t f(X_t, \sigma_t), \\
X_{t+1}^q & = & X_t^q + \Delta_t (f(X_t^q, \sigma_t) + z_t).
\end{array}
\tag{11}
$$

We define the accumulated quantization error as $\delta X_t \equiv X_t^q - X_t$. Taking the the difference of the two equations above, we obtain the recursive equation for $\delta X_{t+1}$

$$
\begin{array}{rcl}
\delta X_{t+1} & = & \delta X_t + \Delta_t \left(f(X_t + \delta X_t, \sigma_t) - f(X_t, \sigma_t) + z_t\right) \\
& = & \delta X_t + \Delta_t \left(\frac{\partial f(X_t, \sigma_t)}{\partial X_t} \delta X_t + R_1(X_t, \sigma_t, \delta X_t) + z_t\right)
\end{array}
\tag{12}
$$

where $R_t \equiv R_1(X_t, \sigma_t, \delta X_t)$ is the Taylor expansion remainder. By expanding the recursive equation (remembering that $\delta X_0 = 0$), we can express $\delta X_{t+1}$ as:

$$
\begin{array}{rcl}
\delta X_{t+1} = & \sum_{\tau=1}^{t} \left(\prod_{k=\tau+1}^{t} \left(I + \Delta_k \frac{\partial f(X_k, \sigma_k)}{\partial X_k}\right) \Delta_\tau (z_\tau + R_\tau)\right) \\
= & \sum_{\tau=1}^{t} \left(\prod_{k=\tau+1}^{t} \mathbb{T}_k \Delta_\tau (z_\tau + R_\tau)\right).
\end{array}
\tag{13}
$$

Here, we use $\mathbb{T}_k \in \mathbb{R}^{\text{latent}} \times \mathbb{R}^{\text{latent}}$ as a shortcut for the expression $I + \Delta_k \frac{\partial f(X_k, \sigma_k)}{\partial X_k}$, noting it is independent of $\{z_t\}_{t=0}^{T-1}$, for all values of $t$. We can now view the accumulated error in the end of the diffusion process ($t = T$), as a sum of three different terms:

$$
\delta X_T = \sum_{\tau=1}^{T} \underbrace{\left(\prod_{k=\tau+1}^{T} \mathbb{T}_k\right) \Delta_\tau (z_\tau - \mathbb{E}z_\tau)}_{s_\tau} + \sum_{\tau=1}^{T} \underbrace{\left(\prod_{k=\tau+1}^{T} \mathbb{T}_k\right) \Delta_\tau \mathbb{E}z_\tau}_{e_\tau} + \sum_{\tau=1}^{T} \underbrace{\left(\prod_{k=\tau+1}^{T} \mathbb{T}_k\right) \Delta_\tau R_\tau}_{r_\tau}.
\tag{14}
$$

Of the three terms, $\sum_t s_t$ describes the effect of the unbiased quantization noise, and is effectively a sum of $i.i.d.$ random variables. $\sum_t e_t$, on the other hand, is deterministic, and describes the accumulated error as a result of biased quantization noise. Finally, $\sum_t r_t$ describes the second-order effects of quantization noise in all steps, which are not independent of each other in practice.

To get a useful expression for the convergence, we will make the following simplifying assumptions:

**Assumption 1.** *The effect of the first order Taylor expansion remainders is negligible, i.e. $\sum_t r_t = 0$.*

**Assumption 2.** *The ratio between the expected value of the norm for the stochastic rounding noise ($\tilde{z}_t$), and the norm of the deterministic noise ($\bar{z}_t$), is bounded in all steps: $\mathbb{E}\|\tilde{z}_t\|^2 = \alpha \|\bar{z}_t\|^2$.*

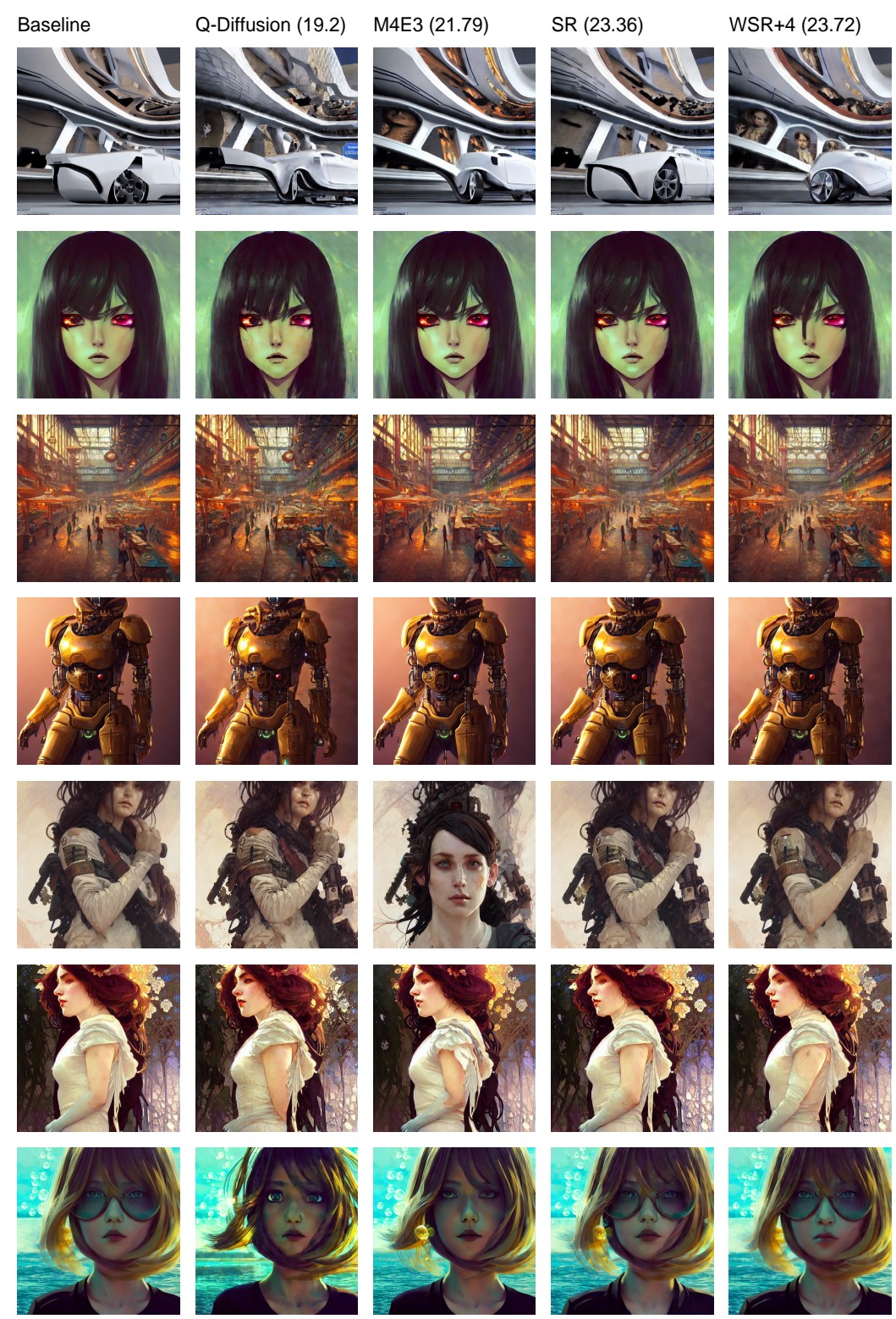

Figure 6: Fixed Seed comparison of generated Stable-Diffusion 1.4 images. Prompts were chosen from the stable-diffusion prompts datasets, **in-order** (no Cherry-picking). The numbers in the top of columns are for the average PSNR over 100 images. Despite unfavorable hyperparameters and harsher quantization, our methods result in images with higher similarity to the full-precision image, and are less likely to exhibit visible errors.

**Assumption 3.** *The norm of each step-quantization noise, when multiplied by step size, is constant:* $\forall i, j \in [1, 2, .., t], \|\Delta_j \bar{z}_j\| = \|\Delta_i \bar{z}_i\|.$

**Assumption 4.** *The correlation matrix between the deterministic quantization noises in each step has a constant, off-diagonal correlation for each pair:* $C = (1 - \bar{c})\mathbb{I} + \bar{c}\mathbb{1}.$

Assumption 1 is an approximation that becomes more accurate for sufficiently small values of $\Delta_t$ (a setting with many diffusion steps), and under the underlying assumption that the function $f(X, \sigma)$ is sufficiently smooth.

Assumption 2 is a simplification of our observation that the MSEs of the quantization noises for both types of quantization are proportional. We empirically examined this observation in SDXL, as we show on Fig. 7. In the case of random, scalar values within (fixed-point) quantization range $([R_{\min}, R_{\max}])$, we can estimate the factor to be:

$$\alpha = \frac{\mathbb{E}_{u,z}\left[Q_{\mathrm{SR}}(u, z) - u\right]^2}{\mathbb{E}_u\left[Q_{\mathrm{R2N}}(u) - u\right]^2} = \frac{\frac{1}{6}}{\frac{1}{12}} = 2, u \sim \mathrm{Uniform}\left(R_{\min}, R_{\max}\right), \quad (15)$$

The prediction in Eq. (15) fits well with the empirical results that were measured for the quantization error of the entire neural network and show $\alpha \sim 4$, for either M4E3, M3E4 or INT8 quantization formats.

Assumption 3 is a simplification of our observation that the empirical norm of $Z_\tau \equiv \Delta_\tau \bar{z}_\tau$ remains somewhat stable, as we show in the right panel of Fig. 7. This is because, as we know from empirical observations (Fig. 2), that quantization noise increase with $t$, while the steps size $(\Delta_\tau)$ decreases. However, in general, the magnitude of the vectors $\{\Delta_\tau \bar{z}_\tau\}_{\tau=0}^T$ is not identical during diffusion, due to changes in $\Delta_\tau$ and changes in the magnitude of the quantization noise, so this is only an approximation.

Assumption 4 simplifies the correlation matrix, highlighting the different impact of off-diagonal and on-diagonal components. As shown in Appendix I, taking $T \to \infty$ (or $\Delta_t \to 0$), will lead to sequential diffusion steps being identical ($f(X_i, \sigma_i) = f(X_j, \sigma_j) \rightsquigarrow C_{ij} = 1$, since $f$ is deterministic). However, realistic settings will rarely align with this asymptotic behaviour. For a more practical result, we use Assumption 4, and take the correlation matrix to be $C = (1 - \bar{c})\mathbb{I} + \bar{c}\mathbb{1}$. Alternatively, we can also think of $\bar{c}$ as the mean of all non-diagonal correlations, which will allow more nuanced correlation matrix and will effectively yield the same results. An example for the correlation matrix in a practical setting can be seen in Fig. 8, where we also measured: $\bar{c} = 0.16$.

Our ultimate goal in this section is to quantify the relative benefit of using a bias-free approach, in comparison with a standard, "greedy" method that prioritizes the reduction of (MSE) errors. To compare these, we will examine two case: first, we will look at $\delta X_T$, the diffusion error in a process with unbiased quantization error $\delta X_t\left(\{\tilde{z}_\tau\}_{\tau=1}^T\right)$, with $\mathbb{E}[\tilde{z}_\tau] = 0$ (i.e. $\tilde{e}_\tau = 0$). We will compare this noise with a deterministic, biased quantization noise $\delta \bar{X}_T\left(\{\bar{z}_\tau\}_{\tau=1}^T\right)$ (i.e. $\bar{s}_\tau = 0$).

**Lemma 1.** *Given Assumptions 1, 2, 3 and 4, the probability of the diffusion quantization error being larger for bias-free quantization, is bounded by:*

$$P\left(\|\delta X_T\| \geq \|\delta \bar{X}_T\|\right) < \quad 28 \exp\left(\frac{-\zeta\omega}{2\alpha}\left(1 - \bar{c} + \bar{c}T\right)\right). \quad (16)$$

*under the condition:* $\|\delta \bar{X}_T\| \geq \frac{1}{6}\left(U + \sqrt{U^2 + 36\sigma^2}\right)$

*with:*

- $\zeta = \frac{\sigma^2}{\sigma^2 + \|\delta \bar{X}_T\|\frac{U}{3}}$, *U being the bound for the L2 norm of* $\{\tilde{z}_\tau\}_{\tau=1}^T$.

- $\sigma$ *is the standard deviation of the accumulated norms of all stochastic quantization errors,* $\sigma^2 = \sum_{\tau=1}^T \mathbb{E}\left\|\left(\prod_{k=\tau+1}^T \mathbb{T}_k\right)\Delta_\tau \tilde{z}_\tau\right\|^2$

- $\omega$ *is the square ratio between the minimal and maximal eigenvalues of* $\left(\prod_{k=\tau+1}^T \mathbb{T}_k\right)$, *i.e.* $\omega \equiv \frac{\lambda_{\min}^2}{\lambda_{\max}^2}$.

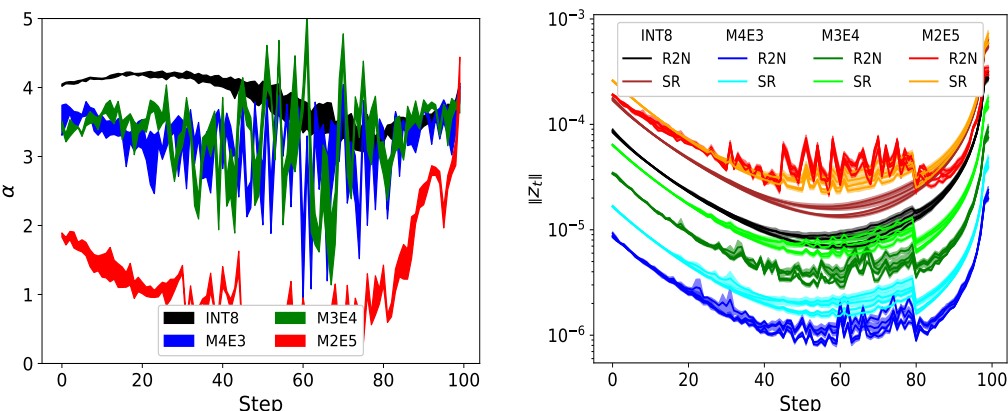

Figure 7: Empirical measurements of the R2N-to-SR norm ratio ($\alpha$) and the quantization noise norm ($\|Z_t\|$). As in Fig. 2, we ran SDXL with 100 diffusion steps, measuring the expected norm of the noise for SR (64 samples), and the norm of the noise for R2N rounding. We used 4 different prompts and 4 different seeds. We reported the min-max margin for $\alpha$ and 1-standard deviation for $\|Z_t\|$. The empirical results (excluding M2E5) support the approximate validity of Assumption 2 and Assumption 3.

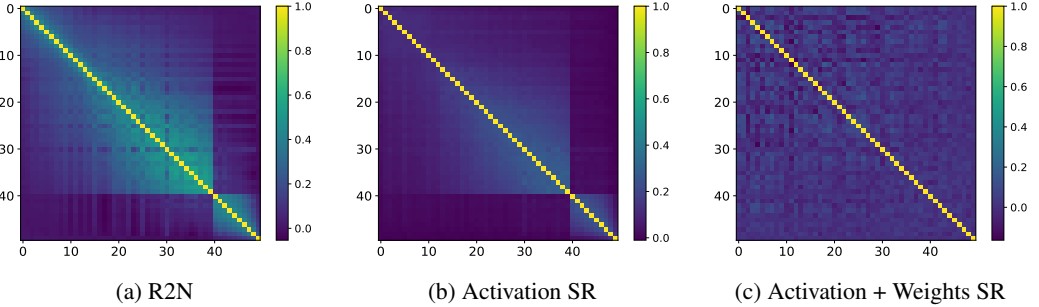

| (a) R2N | (b) Activation SR | (c) Activation + Weights SR |

Figure 8: Correlation matrices between the network's quantization noise on different steps. We used a single prompt, and averaged the results over 4 seeds. When no bias-reduction methods were used, we observed significant correlations between the quantization noise in different steps, as long as we infer with the same neural network (The base model is replaced by the refiner model on step 40). The average off-diagonal correlation for the deterministic quantization noise in this experiment was $\bar{c} = 0.16$. In contrast, the mean off-diagonal correlation was 0.09 and 0.009 with activation SR and activation+weight SR, respectively.

• *The values of both $\alpha$ and $\bar{c}$ originate from Assumptions 2 and 4, respectively.*

*Proof.* With Assumption 1, we can ignore the third term in Eq. (14), and treat $\delta X_T$ as the sum of i.i.d random variables. This allows us to use the extended Vector Bernstein inequality (Minsker, 2011):

$$P\left(\|\delta X_T\| \geq \|\delta \bar{X}_T\|\right) < 28 \exp\left(\frac{1}{2} \frac{-\|\delta \bar{X}_T\|^2}{\sigma^2 + \|\delta \bar{X}_T\| \frac{U}{3}}\right). \tag{17}$$

Eq. (17) is valid under the condition of the Lemma:

$$\|\delta \bar{X}_T\| \geq \frac{1}{6}\left(U + \sqrt{U^2 + 36\sigma^2}\right). \tag{18}$$

By using $\zeta$, we get a more simple form of Eq. (17):

$$P\left(\|\delta X_T\| \geq \|\delta \bar{X}_T\|\right) < 28 \exp\left(\frac{\zeta}{2} \frac{-\left\|\sum_{\tau=1}^{T} \prod_{k=\tau+1}^{T} \mathbb{T}_k \Delta_\tau \bar{z}_\tau\right\|^2}{\mathbb{E} \sum_{\tau=1}^{T} \left\|\prod_{k=\tau+1}^{T} \mathbb{T}_k \bar{z}_\tau\right\|^2 \Delta_\tau^2}\right) \tag{19}$$

The matrix operation $\prod_{k=\tau+1}^{T} \mathbb{T}_k$, accounts for the (first-order) effect of multiple denoising operations over the quantization noise of preceding steps. $\lambda_{\min}$ and $\lambda_{\max}$, the minimal and maximal eigenvalues of all possible matrices, can be used to bound the effect of this operation, in Eq. (20). We now get:

$$P\left(\|\delta X_T\| \geq \|\delta \bar{X}_T\|\right) < 28 \exp\left(\frac{-\zeta}{2} \frac{\lambda_{\min}^2 \left\|\sum_{\tau=1}^{T} \Delta_\tau \bar{z}_\tau\right\|^2}{\lambda_{\max}^2 \sum_{\tau=1}^{T} \Delta_\tau^2 \mathbb{E}\|z_t^2\|}\right) \tag{20}$$

The ratio $\omega \equiv \frac{\lambda_{\min}^2}{\lambda_{\max}^2}$ corresponds to the degree in which the denoiser may be more effective at eliminating deterministic quantization noise, when compared with the elimination of unbiased noise. For a proper bound over the probablity, we must address the worst-case, in which $\omega < 1$.

Next, we use Assumption 2, in which we assume that the ratio between MSE (of the deterministic quantization) and the variance (of the bias-free quantization) is approximately constant: $\mathbb{E}\|z_t\|^2 = \alpha\|\bar{z}_t\|^2$. Using $\alpha$, we get:

$$P\left(\|\delta X_T\| \geq \|\delta \bar{X}_T\|\right) < 28 \exp\left(\frac{-\zeta}{2} \frac{\omega \left\|\sum_{\tau=1}^{T} \Delta_\tau \bar{z}_\tau\right\|^2}{\alpha \sum_{\tau=1}^{T} \Delta_\tau^2 \|\bar{z}_\tau\|^2}\right) \tag{21}$$

If we look at the vectors $Z_\tau \equiv \Delta_\tau \bar{z}_\tau$, and the correlation matrix among these vectors $C_{ij} \equiv \frac{Z_i^T Z_j}{\|Z_i\|\|Z_j\|}$, we can express the probability as:

$$P\left(\|\delta X_T\| \geq \|\delta \bar{X}_T\|\right) < 28 \exp\left(\frac{-\zeta\omega}{2\alpha}\left(1 + 2\frac{\sum_{i=1}^{T} \sum_{j=i+1}^{T} C_{ij}\|Z_i\|\|Z_j\|}{\sum_{i=1}^{T}\|Z_i\|^2}\right)\right). \tag{22}$$

For simplicity, we will take $\|Z_t\|$ to be constant from now on (Assumption 3). With the addition of Assumption 4, Eq. (22) is then simplified to:

$$P\left(\|\delta X_T\| \geq \|\delta \bar{X}_T\|\right) < 28 \exp\left(\frac{-\zeta\omega}{2\alpha}(1 - \bar{c} + \bar{c}T)\right). \tag{23}$$

and the condition will hold whenever Eq. (23) produces a useful probability bound (below 1). This concludes the proof of Lemma 1. $\square$

Eq. (25) gives us a formula for the probability bound, depending on $T, \bar{c}, \alpha$. However, the equation still depends on two additional parameters: $\zeta$, that accounts for the worst-case scenario of large quantization noise, and $\omega$, that account for the worst-case difference in denoising quality, between stochastic and deterministic noise. To get a more intuitive relationship, we will make two additional, optional assumptions:

**Assumption 5.** *The single step norm bound $U$ is negligible when compared with the magnitude of the deterministic quantization error $\left\|\delta\bar{X}_t\right\|$, and specifically, $\frac{U}{\left\|\delta\bar{X}_t\right\|} = \Theta(\frac{1}{T})$.*

**Assumption 6.** *The denoiser effectiveness over the quantization noise (Deterministic or Stochastic) in terms of L2 Norm, is bounded: $\omega \in \Theta(1)$.*

The justification for Assumption 5 is that in our case, $U$ is expected to have a small magnitude: Unlike $\delta\bar{X}_t$ (and $\sigma$), $U$ does not scale with $T$. Furthermore, in a high-dimensional, homogeneous latent space, the norm of a single step quantization noise is highly concentrated (We also saw this empirically, in Fig. 2, for example). This leads us to neglect the effect of $U$, and, when we also take into account the calculated ratio between $\left\|\delta\bar{X}_T\right\|$ and $\sigma$, estimate:

$$\zeta = \frac{\sigma^2}{\sigma^2 + \left\|\delta\bar{X}_t\right\|\frac{U}{3}} = \frac{\frac{\sigma^2}{\left\|\delta\bar{X}_t\right\|^2}}{\frac{\sigma^2}{\left\|\delta\bar{X}_t\right\|^2} + \frac{1}{3}\frac{U}{\left\|\delta\bar{X}_t\right\|}} > \frac{\frac{\alpha}{\zeta\omega}\frac{1}{(1-\bar{c}+\bar{c}T)}}{\frac{\alpha}{\zeta\omega}\frac{1}{(1-\bar{c}+\bar{c}T)} + \frac{1}{3}\frac{k}{T}} = \frac{1}{1 + \frac{\zeta\omega\bar{c}k}{3\alpha}\left(1 + \frac{1-\bar{c}}{\bar{c}T}\right)}. \quad (24)$$

Thus, $\zeta \in \Theta(1)$. In addition, when $\zeta \to 1$, the condition for using the extended Bernstein inequality, Eq. (18), can be written as $\left\|\delta\bar{X}_T\right\| \geq \sigma\left(1 + \Theta\left(\frac{U}{\sigma}\right)\right)$. Then, the condition in Eq. (18) for the validity of Eq. (17) under Assumptions 2, 3, 4 comes down to:

$$\begin{aligned}
\left\|\delta\bar{X}_T\right\| &> \sigma \\
\frac{\left\|\delta\bar{X}_T\right\|^2}{\sigma^2} &> 1 \\
\frac{\zeta\omega}{\alpha}\left(1 + 2\frac{\sum_{i=1}^{T}\sum_{j=i+1}^{T}C_{ij}\|Z_i\|\|Z_j\|}{\sum_{i=1}^{T}\|Z_i\|^2}\right) &> 1 \\
\frac{\zeta\omega}{\alpha}\left(1 - \bar{c} + \bar{c}T\right) &> 1.
\end{aligned} \quad (25)$$

This condition will be met whenever Eq. (23) produces relevant bound ($P < 1$).

As mentioned before, $\prod_{k=\tau+1}^{T}\mathbb{T}_k$ is a matrix operation, that stands for the first-order evolution of quantization noise during multiple denoising iterations. Assumption 6, which concerns the ratio between the maximal and minimal eigenvalues of the operator, is needed since it is not possible to make exact statements regarding the functionality of a neural network over an unknown input. For example, it is possible in theory to have a denoiser which is able to perfectly remove the quantization noise ($\lambda_{\min} = 0$). Nevertheless, we can estimate that the function $f$ is Lipschitz , on account of it being the output of a deep neural network, that was fully trained over a large variety of noisy-inputs. For a Lipschitz constant of $\beta$, we can deduce that the minimal eigenvalue of matrix operation exists, and will be $\lambda_{\min} = \prod_{k=1}^{T}(1 - \Delta_k\beta)$. On the other hand, common sense tells us that the denoising operation will work toward *reducing* the quantization noise, leading us to estimate that the maximal eigenvalue of the operation $\prod_{k=\tau+1}^{T}\mathbb{T}_k$ is $\lambda_{\max} < 1$. Consequently, even in the worst case, we still expect $\omega$ to be non-zero, and $\omega = \frac{\lambda_{\min}^2}{\lambda_{\max}^2} > \lambda_{\min}^2 \rightsquigarrow \omega \in \Theta(1)$, for small enough values of $\beta$. In the general case, we have no reason to assume the denoiser will perform worse over the stochastic noise induced by SR, and can approximate $\omega \sim 1$.

With the optional Assumptions 5 and 6, our concentration bound is valid (Eq. (21)) and is reduced to simply:

$$P\left(\|\delta X_T\| \geq \left\|\delta\bar{X}_T\right\|\right) = O(\exp\left(\frac{-\bar{c}T}{2\alpha}\right)). \quad (26)$$

In conclusion, the primary condition for the bias-free approach to preferable is that the number of diffusion steps $T \in \mathbb{N}$ must be large enough so that $\bar{c}T > 2\alpha$, with $\bar{c}$ being the average correlation between the quantization noise in different diffusion steps, $\alpha$ being the "cost" of the bias-free approach in terms of MSE (Empirically, $\alpha \sim 4$). Once $T$ is large enough, any further increase in the number of diffusion steps will give exponential advantage to the bias-free approach, over methods that use deterministic/biased quantization noise.

## I   ZERO ORDER TAYLOR EXPANSION

For zero order approximation of the correlation between deterministic quantization noises in different steps, we use the fact that for $\gamma = 0$ (deterministic image generation), the step size for $X_t$ is $\Delta_t$, which is defined as $\Delta_t = \sigma_t - \sigma_{t+1}$. For $\gamma > 0$, the step size is modified slightly, resulting in an equation that also depends on $1 + \gamma_t$, but this will not change the result to a significant degree, as long as $\gamma \ll 1$ (any realistic settings).

$$
\begin{aligned}
f\left(X_{t+1}, \sigma_{t+1}\right) &= f\left(X_t + \Delta_t \overrightarrow{D_t}, \sigma_{t+1}\right) \\
&= f\left(X_t, \sigma_{t+1}\right) + \Delta_t \frac{\partial f(X_t, \sigma_{t+1})}{\partial X_t} \cdot \overrightarrow{D_t} + \mathcal{O}\left(\Delta_t^2\right) \\
&= f\left(X_t, \sigma_t - \Delta_t\right) + \Delta_t \frac{\partial f(X_t, \sigma_{t+1})}{\partial X_t} \cdot \overrightarrow{D_t} + \mathcal{O}\left(\Delta_t^2\right) \\
&= f\left(X_t, \sigma_t\right) - \Delta_t \frac{\partial f(X_t, \sigma_t)}{\partial \sigma_t} + \Delta_t \frac{\partial f(X_t, \sigma_t - \Delta_t)}{\partial X_t} \cdot \overrightarrow{D_t} + \mathcal{O}\left(\Delta_t^2\right)
\end{aligned}
\tag{27}
$$

When the number of total diffusion steps is high, $\Delta_t$ approach 0 and the model outputs are identical in sequential steps, $f\left(X_{t+1}, \sigma_{t+1}\right) = f\left(X_t, \sigma_t\right)$. In this case, it is evident that the (deterministic) quantization noises in both steps must be identical as well, i.e. $C_{t,t+1} = 1$.

## J   MATCHING QUANTIZATION NOISE TO DISCRETE EULER NOISE

In section 4.1, we came up with a simplistic model for quantization noise, as expressed in Eq. (28):

$$
f_Q\left(X_t, \sigma_t\right) = f_{\text{FP32}}\left(X_t, \sigma_t\right) + z_t \quad, z_t \sim \mathcal{N}\left(\bar{\mu}_t, \mathbb{I}\sigma_{Q,t}\right).
\tag{28}
$$

$\sigma_{Q,t}$ denotes the standard deviation of quantization noise, and was assumed to be i.i.d. for all scalar values in the output. Based on former notations, we can see that

$$
\sigma_{Q,t} = \sqrt{r_t^2 - \bar{\mu}_t^2}
\tag{29}
$$

In this section, our goal is to adjust the Discrete Euler Scheduler (Algorithm 1), to get a result that is consistent with the existing algorithm, but does not explicitly include an addition of noise– the noise will be implicitly added by the network quantization (Eq. (28)).

We will take the assumption that our other methods, for de-biasing the network, were successful, allowing us to take $\bar{\mu}_t = 0$, in the context of Eq. (28). This means that in theory, the quantization has the effect of adding white noise, with a standard deviation of $\sigma_{Q,t}$.

For quantized diffusion, the model output is only added to the existing latent space after being multiplied by $\Delta_t$ (again, from Algorithm 1). The relevant addition of noise, in this case, is $z_t \cdot \Delta_t$, which we would like to be identical to an added noise in a non-quantized setting, where $\gamma > 0$. Forcing the variance in both cases to be identical is a bit more tricky than expected, since both terms also depend on $\gamma$, which we would like to keep identical, to be able to claim that the algorithms are equivalent (same added noise, same noise schedule, same size of steps). Nevertheless, we can extract $\gamma$ that comply with these conditions:

$$
\begin{aligned}
\text{Var}\left(\sqrt{\hat{\sigma}_t^2 - \sigma_t^2}\,\epsilon_t\right) &= \text{Var}(z_t \cdot \Delta t) \\
\left(\hat{\sigma}_t^2 - \sigma_t^2\right)\text{Var}\left(\epsilon_t\right) &= \left(\Delta t\right)^2 \text{Var}\left(z_t\right) \\
\hat{\sigma}_t^2 - \sigma_t^2 &= \sigma_{Q,t}^2 \left(\sigma_{t+1} - \hat{\sigma}_t\right)^2 \\
\left(\gamma^2 + 2\gamma\right)\sigma_t^2 &= \sigma_{Q,t}^2 \left(\sigma_{t+1} - (\gamma+1)\sigma_t\right)^2 \\
\left(\gamma^2 + 2\gamma\right) &= \sigma_{Q,t}^2 \left(\frac{\sigma_{t+1}}{\sigma_t} - (\gamma+1)\right)^2
\end{aligned}
\tag{30}
$$

We define $r \equiv \frac{\sigma_{t+1}}{\sigma_t}$, to get:

$$
\begin{aligned}
\sigma_{Q,t}^2 \left(r^2 - 2r\left(\gamma+1\right) + \left(\gamma^2 + 2\gamma + 1\right)\right) &= \left(\gamma^2 + 2\gamma\right) \\
\left(\gamma^2 + 2\gamma\right) - \sigma_{Q,t}^2 r^2 + 2r\sigma_{Q,t}^2\left(\gamma+1\right) - \sigma_{Q,t}^2\left(\gamma^2 + 2\gamma + 1\right) &= 0 \\
\gamma^2\left(1 - \sigma_{Q,t}^2\right) + 2\gamma\left(1 + (r-1)\sigma_{Q,t}^2\right) - \sigma_{Q,t}^2\left(r^2 - 2r + 1\right) &= 0
\end{aligned}
\tag{31}
$$

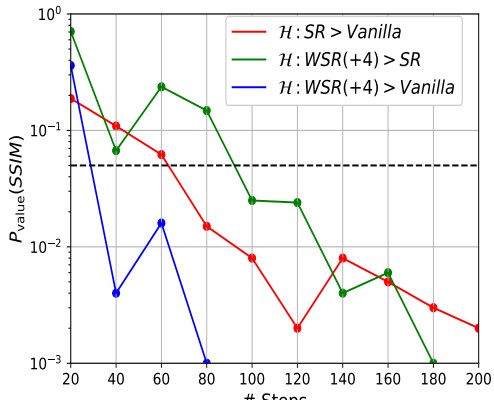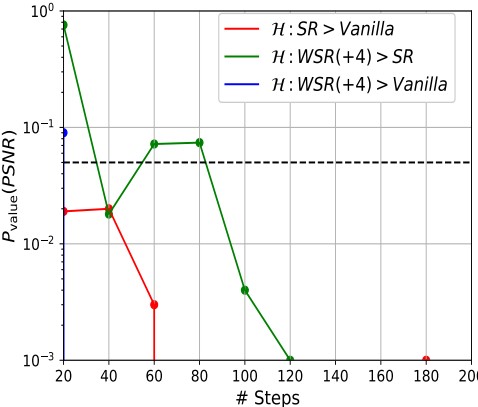

Figure 9: Statistical Significance of the main results. This figure show the p-values (calculated via 1-sided t-test), corresponding to the hypotheses that SR and WSR improve the SSIM (Left) or the PSNR (Right) of generated images when compared with their full precision counterparts. This analysis is based on the results presented in figure Fig. 3 (Right).

with the solution:

$$\gamma = \frac{\sqrt{1 + \sigma_{Q,t}^2 \left(1 - \left(\frac{\sigma_{t+1}}{\sigma_t}\right)^2\right)} + \left(1 - \frac{\sigma_{t+1}}{\sigma_t}\right)\sigma_{Q,t}^2 - 1}{\left(1 - \sigma_{Q,t}^2\right)} \tag{32}$$

If the noise is small enough $\sigma_Q \ll 1$, we get:

$$\begin{aligned}
\gamma &\simeq \frac{1 + \frac{1}{2}\sigma_{Q,t}^2\left(1 - \left(\frac{\sigma_{t+1}}{\sigma_t}\right)^2\right) + \left(1 - \frac{\sigma_{t+1}}{\sigma_t}\right)\sigma_{Q,t}^2 - 1}{\left(1 - \sigma_{Q,t}^2\right)} \\
\gamma &= \sigma_{Q,t}^2\left(1 - \frac{\sigma_{t+1}}{\sigma_t}\right)\left[\frac{1}{2}\left(1 + \frac{\sigma_{t+1}}{\sigma_t}\right) + 1\right] \\
\gamma &= \frac{1}{2}\sigma_{Q,t}^2\left(1 - \frac{\sigma_{t+1}}{\sigma_t}\right)\left(3 + \frac{\sigma_{t+1}}{\sigma_t}\right) \\
\gamma &\simeq \frac{1}{2}\sigma_{Q,t}^2\left(1 - \frac{\sigma_{t+1}}{\sigma_t}\right)\left(3 + \frac{\sigma_{t+1}}{\sigma_t}\right)
\end{aligned} \tag{33}$$

## K    ROBUSTNESS TO CHANGES IN HYPER-PARAMETERS

All our work in prior sections was based on the default parameters of the Stable-Diffusion XL pipeline, to ensure the relevancy of our method to practical settings. Nevertheless, we would still like our method to be functional for non-default settings. With this in mind, we proceeded to study the effect of changed parameters over our main results– and see how the SR/WSR methods hold out for different sets of hyperparameters.

The first likely candidates to have a non-default are the resolution of the generated image, and the Classifier-Free Guidance (CFG) scale. The guidance scale determines the strength in which the diffusion process adheres to the condition (as dictated by the prompt), and is therefore changed often to balance the likelihood of the generated image and how much it matches the desired prompt. In Fig. 10, we added the PSNR/SSIM results for image resolutions of $640 \times 1536$ or $1344 \times 768$ (The default was $1024 \times 1024$), and guidance scaling with values of $3.0$ and $7.0$ (Our default value for guidance scale so far was $5.0$).

Another key change we can expect in practical settings is that some users may prefer sampling images with different schedulers. This is a more substantial change from our perspective, as our theoretical derivations in this work relied on the Euler Discrete Scheduler (Algorithm 1. Nonetheless, when testing the UniPC Multi-Step Scheduler and the Heun Discrete Scheduler (also in Fig. 10), we see that

SR and WSR are still highly beneficial. When combined with the results for varying guidance scales, we conclude that our methods are robust and can be used in a large variety of settings, although we did see that the overall image similarity is effected by the guidance scales/ schedulers. We also observed an anomaly for the UniPC Multi-Step Scheduler, where the image similarity unexpectedly dropped for large numbers of steps ($> 140$).

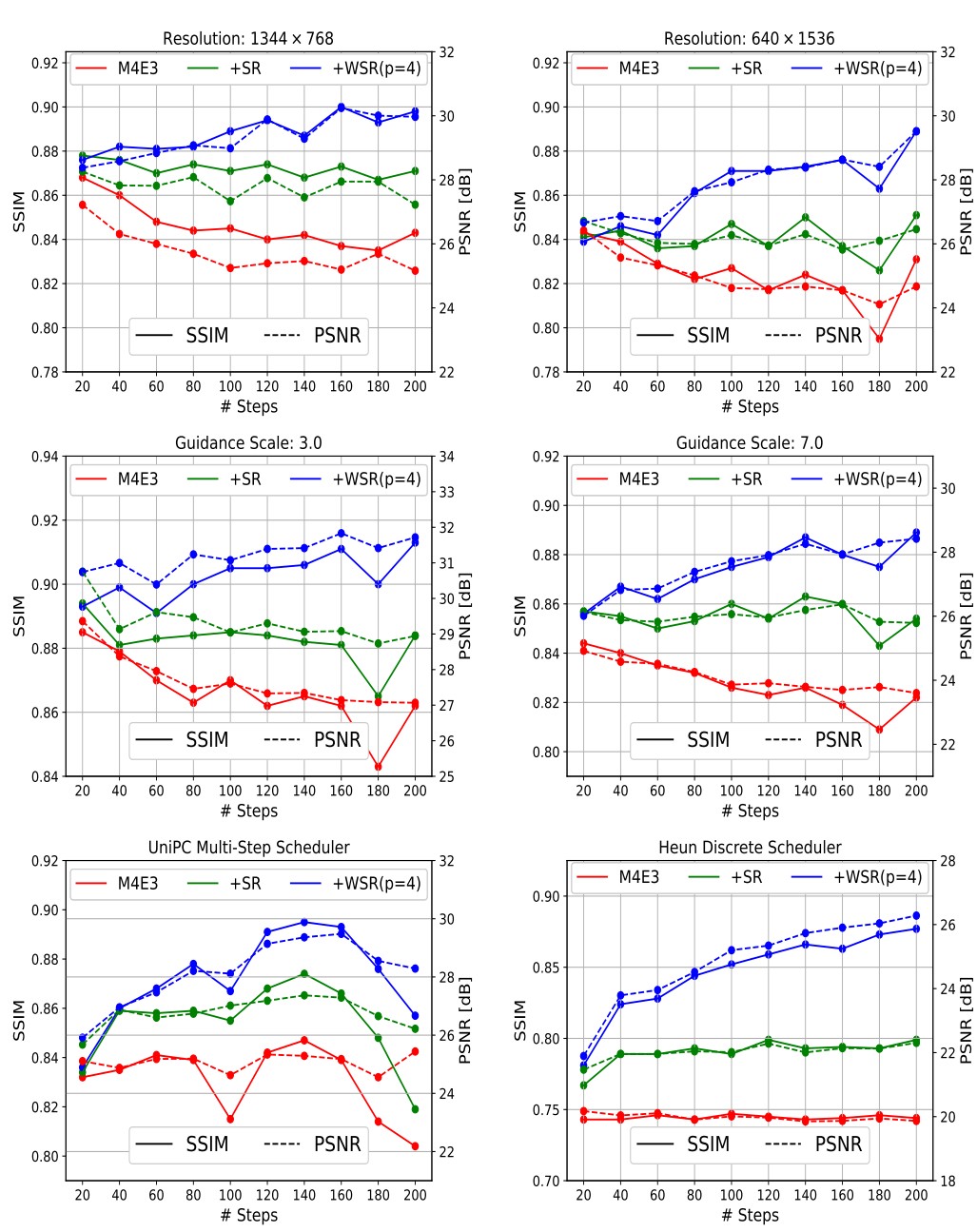

Figure 10: Robustness of SR and WSR, when tested with non-default hyperparameters (Image Resolution, Guidance Scales or Schedulers). We measured both SSIM and PSNR for all experiments. In all cases, stochastic rounding for activations improves image similarity (compared with the FP32 baseline), and stochastic weights improve image similarity further.

## L    REMOVAL OF BIAS ALSO REMOVES CORRELATIONS

Suppose $g_\zeta(X)$ is an unbiased function (e.g., 'quantizer') with random ("quantization noise") variable $\zeta$ which is independently drawn from $X$. The variable $X$ and $g_\zeta(X)$ can be a single scalar quantity (e.g., weight or activation) and its corresponding SR-quantized value, or a neural network output and its corresponding value after we used SR to quantize its weights and/or activations. For simplicity, we assume $X$ and $g_\zeta(X)$ are both scalar—everything can be straightforwardly generalized to vector-valued quantities.

Since it is unbiased, its ('quantization') error $\delta_\zeta(X) \equiv X - g_\zeta(X)$ has zero mean for a given $X$, i.e. $\mathbb{E}[\delta_\zeta(X)|X] = 0$. Therefore, it is also unbiased if we have randomness on $X$, since

$$\mathbb{E}\delta_\zeta(X) = \mathbb{E}\mathbb{E}[\delta_\zeta(X)|X] = 0.$$

Moreover, the error is uncorrelated with $X$

$$\mathbb{E}\left[(X - \mathbb{E}X)(\delta_\zeta(X) - \mathbb{E}\delta_\zeta(X))\right]$$
$$= \mathbb{E}\left[\mathbb{E}\left[(X - \mathbb{E}X)\delta_\zeta(X)|X\right]\right]$$
$$= \mathbb{E}\left[(X - \mathbb{E}X)\mathbb{E}\left[\delta_\zeta(X)|X\right]\right]$$
$$= 0.$$

Lastly, if we are given a sequence of $\{X_t\}_{t=1}^T$ and $\{\zeta_t\}_{t=1}^T$ where $\zeta_t$ is independent of $\{X_{t'}\}_{t'=1}^t$ and $\{\zeta_{t'}\}_{t' \neq t}$ then the error is uncorrelated between different times

$$\mathbb{E}\left[\left(\delta_{\zeta_t}(X_t) - \mathbb{E}\delta_{\zeta_t}(X_t)\right)\left(\delta_{\zeta_{t+\tau}}(X_{t+\tau}) - \mathbb{E}\delta_{\zeta_{t+\tau}}(X_{t+\tau})\right)\right]$$
$$= \mathbb{E}\left[\delta_{\zeta_t}(X_t)\delta_{\zeta_{t+\tau}}(X_{t+\tau})\right]$$
$$= \mathbb{E}\left[\mathbb{E}\left[\delta_{\zeta_t}(X_t)\delta_{\zeta_{t+\tau}}(X_{t+\tau})|X_t, X_{t+\tau}, \zeta_t\right]\right]$$
$$= \mathbb{E}\left[\delta_{\zeta_t}(X_t)\mathbb{E}\left[\delta_{\zeta_{t+\tau}}(X_{t+\tau})|X_{t+\tau}\right]\right]$$
$$= 0$$

## M    IMAGE SIMILARITY OF COMMON BASELINES

In previous sections, we briefly mentioned the SSIM scores of images generated using FP16 and BF16 neural networks, when compared with the corresponding images of full-precision networks. In this section, we will expand over these results, with the hope of giving full context to the results presented in the paper.

The three most common data types in use for diffusion models are full-precision (FP32), half-precision (FP16), and brain-float (BF16), which have the mantissa-exponent allocation of M23E8, M10E5 and M7E8, respectively. To get a proper baseline, we will use the publicly available SDXL models (FP32 for full-precision and brain-float, or FP16 for half precision), and cast the model to a pytorch's data type (*torch.float32*, *torch.float16* or *torch.bfloat*). The SSIM and PSNR of the images generated by each format (compared with FP32), are presented in figure Fig. 11, next to the SSIM/PSNR achieved using the M4E3 format (Fig. 3), with or without the addition of stochastic rounding (weights and activations). Surprisingly, our results show that the methods presented in this paper are sufficient to close the gap with the BF16 datatype completely, when using longer diffusion processes (more than 180 steps).

## N    LIMITATIONS

This paper proposes a new method to quantized diffusion models, and relies on simulation of hardware that is not, at the current moment in time, available for commercial/ academic use. Estimations of performance for different components were made without exact knowledge of the architecture in which they will be implemented. While we have consulted hardware architects and designers for the purpose of this work, our assessment regarding hardware costs remains crude, and large differences in implementations are expected depending on the hardware our proposed features will be implemented on.

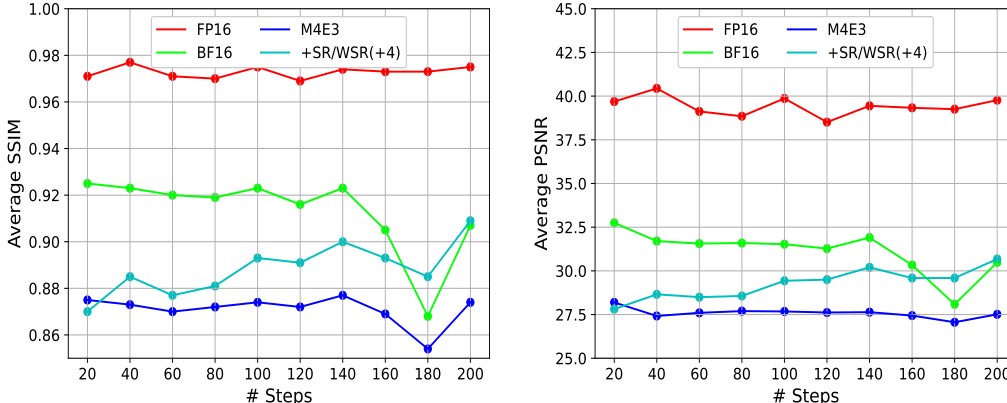

Figure 11: The Image Similarity achieved by our methods, compared with common-baselines. While the FP16 variant of the SDXL model results in images that are practically indistinguishable from the FP32 variant, using brain-float causes noticeable changes in the generated images. For the longer diffusion processes, our methods can be used to generated images with the same amount of distortion that was deemed acceptable for brain-float.

Our hardware simulation performed explicit quantization for all values before Matrix multiplication, Convolutions, and Linear operations. While this may sufficient for enhancing performance in most hardware settings, we do expect hardware implementation to perform quantization after GeMM operations as well, so that activations, normalization, and addition operations can be performed in low precision as well. From prior experience, we don't expect major issues to occur due to quantization in these parts and therefore did not include them in our simulation.

In our analysis for the effect of error and bias on the diffusion process we have used a Taylor expansion of the U-Net output (**??**), to establish the behaviour in the regime where $\Delta t$ approaches zero. The validly of this approximation relies on an implicit assumption that $\frac{\partial f(X_t, \sigma_{t+1})}{\partial X_t}$, and more importantly, $\frac{\partial f(X_t, \sigma_t)}{\partial \sigma_t}$, are bounded $\left( < \frac{1}{\Delta t} \right)$. While this is a reasonable assumption to take in the case of SDXL, it will not hold in the case where the diffusion process is done by multiple neural networks, that were trained to work on distinct time-steps. Therefore, we do not expect our method to be effective in this setting.

The main drawback of using similarity as a goal is that any change in details in the generated image will be considered a mistake, even when it appears to be preserving the image quality, and is not visibly affecting the distribution of the generated images.

Besides the SSIM criteria, our main quantitative results rely on the FID criteria that appear to be ill-suited for the task of evaluating images generated with SDXL, even when the number of generated images is large (30K).

## O  VISUAL EXAMPLES

Figs. 12 and 13 contain comparisons between step size and quantization methods for a fixed prompt and seed. For more image examples, see the supplementary material of this work.

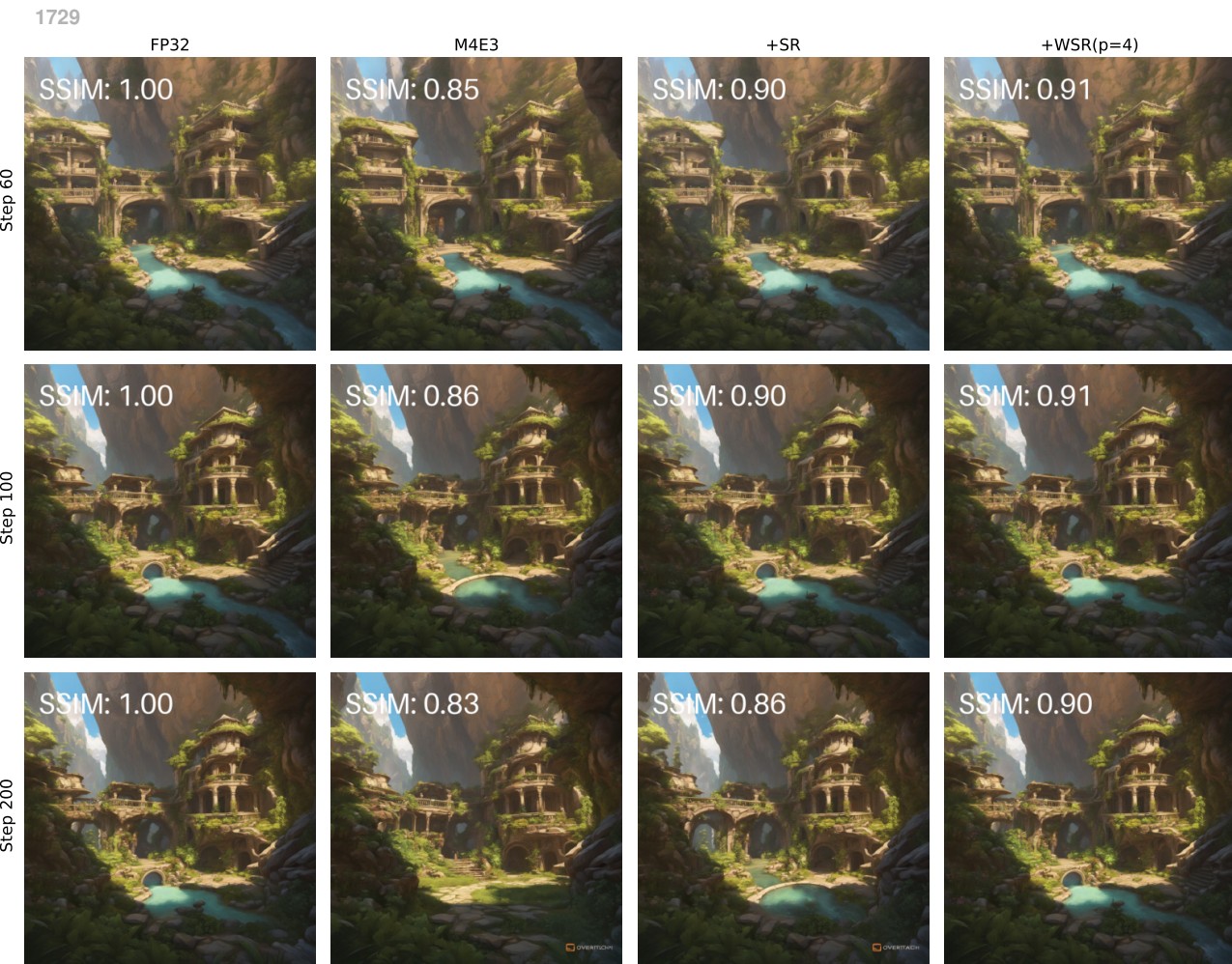

Figure 12: Generated images comparison. Stable diffusion prompt. On a surface level comparison, all images appear to be similar, and of similar quality. Implementing our method results in images that share (more of) the fine-details that are found in the baseline image (like the water structure in this example).

Table 4: CLIP and Inception score measurements of 30K images generated by quantized diffusion models IS and CLIP measurements were generally insufficient for the task of distinguishing between the output of the different models.

| Steps | 20 | | 50 | | 100 | | 200 | |
|---|---|---|---|---|---|---|---|---|
| | CLIP↑ | IS↓ | CLIP↑ | IS↓ | CLIP↑ | IS↓ | CLIP↑ | IS↓ |
| FP16 | 0.3204 | 1.135 | 0.3207 | 1.138 | 0.3204 | 1.139 | 0.3205 | 1.140 |
| FP8 | 0.3204 | 1.132 | 0.3208 | 1.136 | 0.3206 | 1.138 | 0.3206 | 1.138 |
| +SR | 0.3200 | 1.132 | 0.3210 | 1.137 | 0.3209 | 1.139 | 0.3208 | 1.139 |
| +p=4 | 0.3200 | 1.137 | 0.3208 | 1.140 | 0.3207 | 1.141 | 0.3207 | 1.142 |
| TRT FP16 | 0.3200 | 1.133 | 0.3205 | 1.141 | 0.3205 | 1.142 | 0.3204 | 1.142 |
| TRT INT8 | 0.3184 | 1.128 | 0.3125 | 1.088 | 0.2024 | 1.004 | 0.1973 | 1.002 |

1782
1783
1784
1785
1786
1787
1788
1789
1790
1791

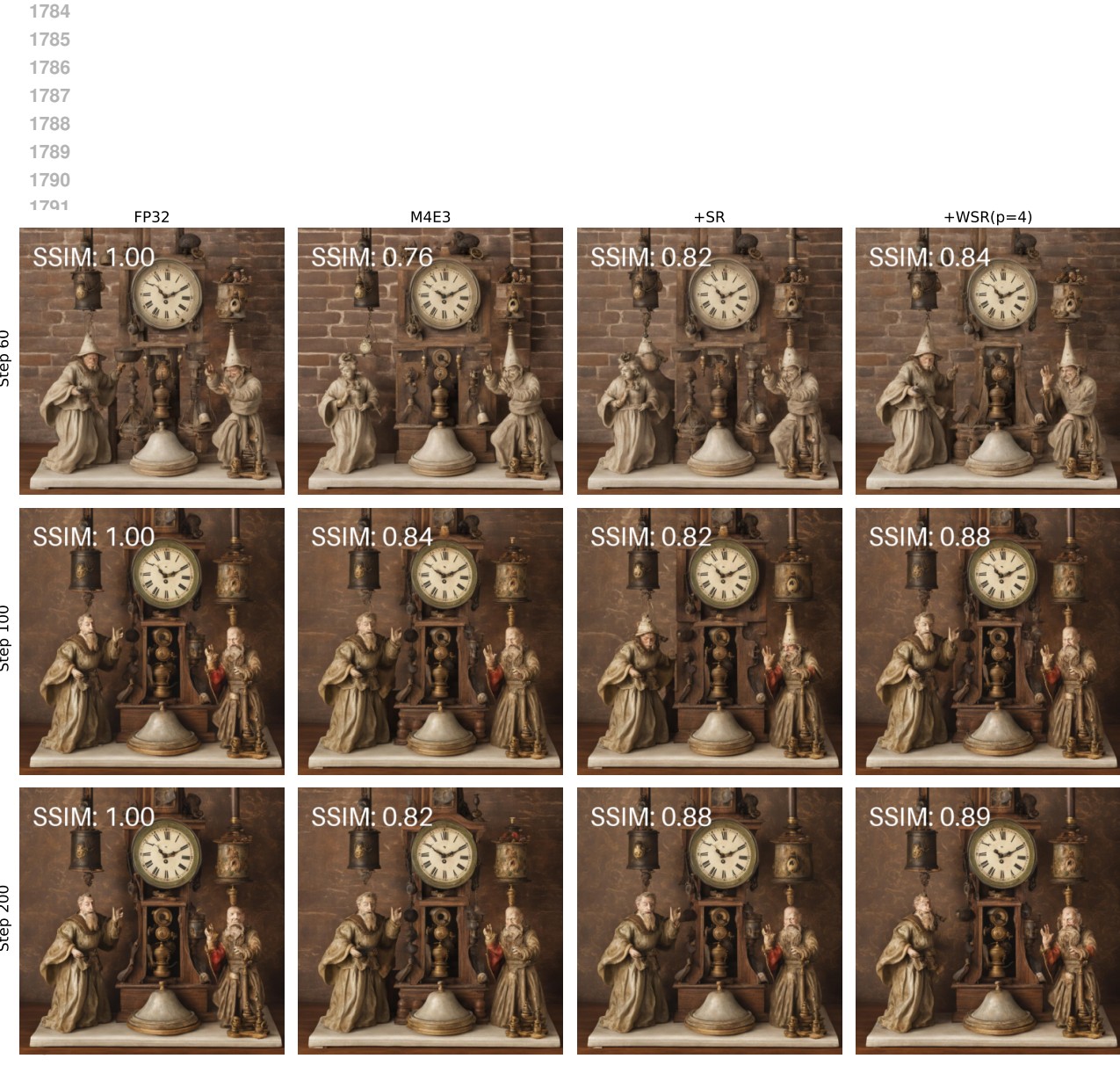

Figure 13: Generated images comparison. MS-COCO prompt. Like the previous example, ranking the quality of images that were generated by different methods is not trivial. Small details that appear in the full-precision images are more likely to be included in images generated using our methods. In the case of MS-COCO dataset, our methods resulted images that achieved better FID in comparison with the images in the dataset.

1822
1823
1824
1825
1826
1827
1828
1829
1830
1831
1832
1833
1834
1835

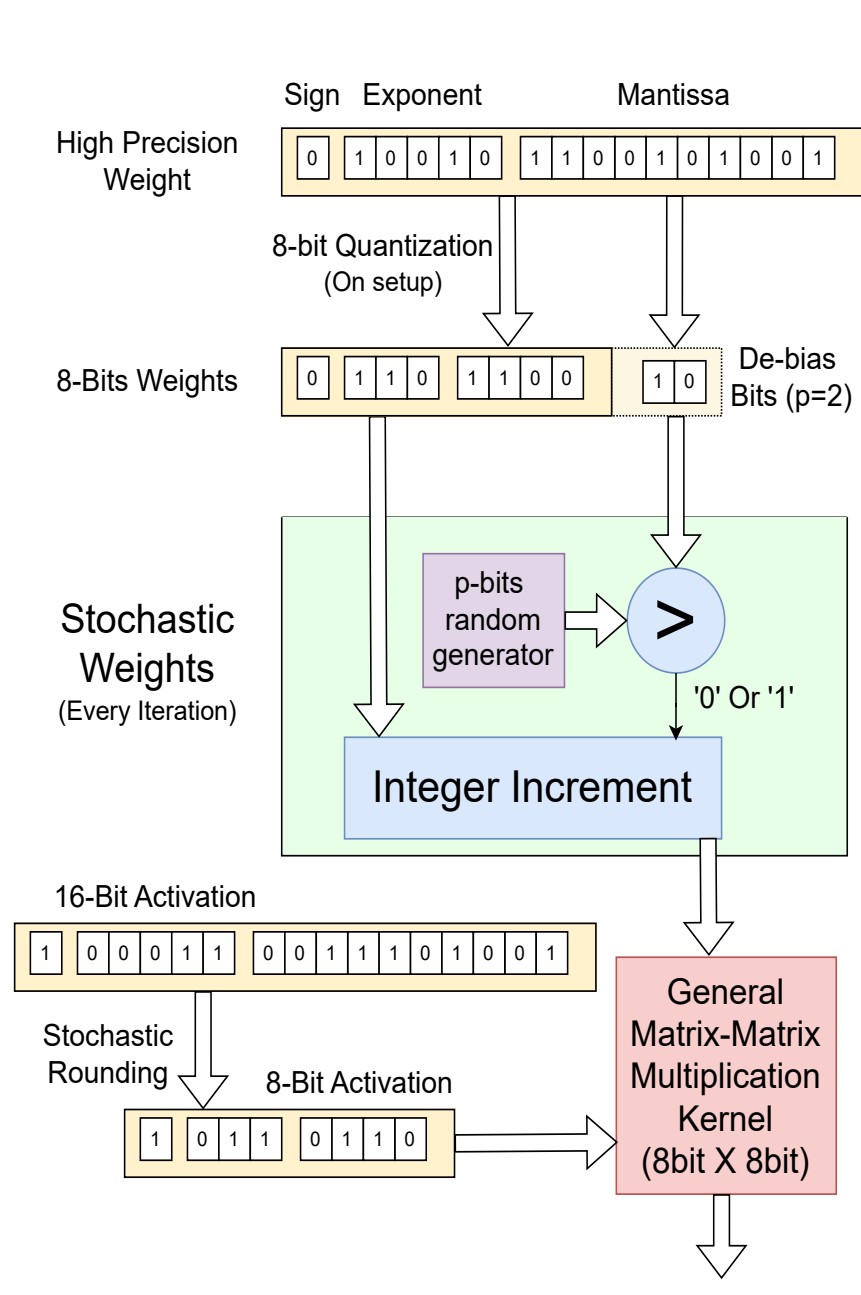

Figure 14: Visualization of Stochastic Rounding and Stochastic Weights

