# OpenReview forum: "De-biasing Diffusion: Data-Free FP8 Quantization of Text-to-Image Models with Billions of Parameters"
_ICLR.cc/2025/Conference — Submitted to ICLR 2025_

### Official Review · Reviewer_6A3r · 2024-11-01

**Soundness:** 3
**Presentation:** 3
**Contribution:** 2
**Rating:** 5
**Confidence:** 4

**Summary:**

This manuscript proposes a FP-based Diffusion Model quantization scheme. Specifically, the paper uses Stochastic Rounding, originally used for training, to assist the quantization calibration process. The technique uses PSNR as a measure to try and achieve calibration where images produced by a quantized denoising U-Net/Transformer are very similar to that of the full precision model. This approach is applied to SDXL and Flux.1.

**Strengths:**

- Floating point quantization for diffusion models is positive direction.
- The paper is handling the larger models, SDXL and Flux.1.
- The approach is generally intuitive and easy to understand.
- This paper builds on top of PTQD [1] in adjusting the denoising schedule, which is good as not a lot of diffusion quantization paper adopt its approach.
- Figure 2 is effective.

**Weaknesses:**

- The primary similarity metric used by the authors for calibration is PSNR. They cite the SDXL [2] paper which states that FID has limitations due to Mean-Opinion-Scores. This is usually because the generates images are compared to real images (FID-GT). However, [3] propose FID-FP32, a variant of FID which compares the distribution of generated images from the quantized model, against those of the full precision model (which is treated as the baseline). If FID-32 is 0, the distribution of images from the quantized model is the same as that of the full precision model. Since this metric discards the weaknesses of the original FID, it is arguably a better metric for the objective than PSNR for what the authors aim to achieve, yet this paper does not seem to be aware of it.
- Most of the experimental results generate too few images (100 or 1k, e.g., Fig 1, Fig 3, Tab 2) to provide trustworthy and meaningful results.
- Line 463-464 "the minimal recommended size" this is incorrect. See https://huggingface.co/docs/diffusers/en/using-diffusers/sdxl which states "anything below 512x512 is not likely to work".
- The paper only considers W8 precision, whereas other methods consider W4 [3].

**Questions:**

See above weaknesses

References:

[1] He, Yefei, et al. "Ptqd: Accurate post-training quantization for diffusion models." Advances in Neural Information Processing Systems 36 (2024).

[2] Podell, Dustin, et al. "Sdxl: Improving latent diffusion models for high-resolution image synthesis." arXiv preprint arXiv:2307.01952 (2023).

[3] Tang, Siao, et al. "Post-training quantization with progressive calibration and activation relaxing for text-to-image diffusion models." arXiv preprint arXiv:2311.06322 (2023).

---

> ### Author Response · Authors · 2024-11-20
> **Response to Reviewer 6A3r**
>
> > The primary similarity metric used by the authors for calibration is PSNR. They cite the SDXL [2] paper which states that FID has limitations due to Mean-Opinion-Scores. This is usually because the generates images are compared to real images (FID-GT). However, [3] propose FID-FP32, a variant of FID which compares the distribution of generated images from the quantized model, against those of the full precision model (which is treated as the baseline). If FID-32 is 0, the distribution of images from the quantized model is the same as that of the full precision model. Since this metric discards the weaknesses of the original FID, it is arguably a better metric for the objective than PSNR for what the authors aim to achieve, yet this paper does not seem to be aware of it.
>
> Many thanks for this comment. We were not aware of this metric (It is a 2025 publication), and have gladly added it to the paper. This is a neat way to measure similarity while focusing on key features, as perceived by the Inception model. When evaluated over our COCO images (30K each), using pytorch-fid (dims=2048), the results are as follow:
>
> |     |   fp32 |   fp16 |   bf16 |   M4E3 |   +SR |   +WSR (p=4) |    TRT |
> |----:|-------:|-------:|-------:|-------:|------:|-------------:|-------:|
> |  20 |      0 |   0.28 |   0.65 |   **1.14** |  1.19 |         1.16 |   2.16 |
> |  50 |      0 |   0.18 |   0.88 |   1.06 |  1.02 |         **0.94** |  39.61 |
> | 100 |      0 |   0.17 |   0.56 |   1.05 |  0.99 |         **0.82** | 465.96 |
> | 200 |      0 |   0.17 |   0.81 |   1.05 |  0.97 |         **0.72** | 514.48 |
>
> The new results were added to the table in Figure 5 (d), next to the SSIM results. FID-FP32 is explained (with reference to the QDiffBench paper), in appendix E (renamed to: “Similarity Measurements via SSIM and FID-32”). Note the TensorRT results have a different FID baseline. We also used pytorch metrics to corroborate these values.
>
> As expected, the FID-FP32 values are relatively small: The datasets we compare have much stronger likelihood than what we usually measure with FID-GT. In regard to the methods proposed by this paper, the FID-FP32 is consistent with the PSNR/SSIM scores: SR and WSR are beneficial for $T>20$, with WSR achieving almost as good a score as a naive bf16 quantization.
>
> > Most of the experimental results generate too few images (100 or 1k, e.g., Fig 1, Fig 3, Tab 2) to provide trustworthy and meaningful results.
>
> The MS-COCO experiment uses 30K images (per FID value), not 1K. Other experiments used $100$ images (scanned over 10 different values of T), but the results were robust (i.e. the results did not significantly change with more images), as indicated by the very low p-values (Figure 9). Specifically, our methods show a clear statistically significant benefit ($p<0.05$) for each number of diffusion steps (except for $T=20$), which indicates we used enough images to verify our advantage.
>
> > Line 463-464 "the minimal recommended size" this is incorrect. See https://huggingface.co/docs/diffusers/en/using-diffusers/sdxl which states "anything below 512x512 is not likely to work".
>
> These statements are not contradictory: 512x512 is likely to work, but it is not recommended, and will result in lower quality images (Special models were fine-tuned to address this domain [1]). However, we agree that the term “minimal” is problematic in relation to two-dimensional sizes and have removed this claim.
>
> [1] https://huggingface.co/hotshotco/SDXL-512

---

> > ### Comment · Reviewer_6A3r · 2024-11-23
> > **Maintaining Score**
> >
> > I thank the author for their rebuttal. After taking the time to thoroughly examine it as well as the other reviewers' concerns, I choose to maintain my score of 5 for borderline reject. The weaknesses do not outweigh the strengths, and I found the rebuttal did not accurately address my concerns.

---

> > > ### Author Response · Authors · 2024-11-24
> > >
> > > We thank the reviewer for the time and effort invested in this review and the re-examination. We will be happy to hear what the remaining/new issues are, to help us improve our paper.

---

### Official Review · Reviewer_Xmw8 · 2024-11-02

**Soundness:** 2
**Presentation:** 2
**Contribution:** 2
**Rating:** 5
**Confidence:** 4

**Summary:**

This work proposes stochastic rounding of weights and activations to mitigate the bias incurred by quantization in the iterative denoising process. The introduced approach is evaluated on fp8 quantization of modern diffusion models - SDXL and FLUX.

**Strengths:**

* Bias-variance decomposition of quantization error makes sense and provides some useful insights about the possible solution to reduce the quantization error,

* The superiority of the proposed stochastic rounding for weights and activations compared to naive rounding-to-nearest is validated on several modern architectures and inference setups.

**Weaknesses:**

* The overall novelty is limited. Stochastic rounding is an established technique in various applications [1, 2] and the current work adopts it for diffusion models.

* The evaluation setup is not very compelling. Unless the drop of generation quality is very significant, metrics such as FID do not reflect it. PSNR/SSIM between the generations produced by the original model and quantized one may be not a proper measure, as a compressed model can generate samples different from the teacher, yet of high quality. I would suggest adding user preference study at least for some cases as a reliable assessment. Specifically, given two images an assesor should tell, whether one image is better than other (or both are equal in terms of quality).

* WSR(p=4) increases amount of memory used for model storage by 50% relative to fp8. I would suggest to check whether mixed fp8/fp16 quantization is still inferior in terms of quality.

**Minor**

Line 84: PTDQ -> PTQD

---
[1] Gupta, Suyog, et al. "Deep learning with limited numerical precision." International conference on machine learning. PMLR, 2015.
[2] Wang, Naigang, et al. "Training deep neural networks with 8-bit floating point numbers." Advances in neural information processing systems 31 (2018).

**Questions:**

* The paper focuses on fp8 quantization, where performance drops relative to uncompressed model are not very pronounced. Would be interesting to see whether the conclusions are valid for the case of more aggressive quantization (say W4/A8) - i.e one should focus on reducing the bias, rather than the variance term. I suggest having at least one experiments with stronger compression to showcase the difference between RTN performance and SR.

* Noticeable decrease of FID metric relative to uncompressed model given high PSNR (small difference between the output of original and quantized model) looks suspicious. Do you have any explanation for this phenomenon? Do you adopt the same random seed for generation with compressed and original model? How robust is the FID value to the change of seed?

* What is the sampler used for image generation? Is it the default sampler in diffusers config or DDIM?

---

> ### Author Response · Authors · 2024-11-20
> **Response 1/2 to Reviewer Xmw8**
>
> > The overall novelty is limited. Stochastic rounding is an established technique in various applications [1, 2] and the current work adopts it for diffusion models.
>
> The main novelty of our paper is pinpointing quantization bias as the key issue in diffusion inference tasks, and showing the implications of this. Stochastic rounding (SR) is a tool for reducing bias, but it increases the quantization error (MSE), so in many cases using SR actually degrades the accuracy. For example, this happens often for weight and activations quantization in the forward pass in classification models (e.g. see Fig 1b in Chmiel et al. 2022, cited in our paper). Thus, the fact that SR has been used for other, vastly different applications (e.g., [1,2] quantized the gradient-based training process of image classification models) does not contradict the novelty of our findings. We have explicitly used stochastic rounding because it is a well-established method, allowing us to benefit from the already existing implementation in software and hardware.
>
> > The evaluation setup is not very compelling. Unless the drop of generation quality is very significant, metrics such as FID do not reflect it. PSNR/SSIM between the generations produced by the original model and quantized one may be not a proper measure, as a compressed model can generate samples different from the teacher, yet of high quality. I would suggest adding user preference study at least for some cases as a reliable assessment. Specifically, given two images an assesor should tell, whether one image is better than other (or both are equal in terms of quality).
>
> As a quantization paper, our main goal was to produce models that make as little as possible changes to the non-quantized model’s output, with the underlying assumption that the output of the non-quantized model was good enough for the user. Having high PSNR images (high-similarity), means we are close to the original images, with the FID giving us some assurance that the changes maintain a reasonable coverage of the distribution. On the other hand, low PSNR images would mean that we have strayed far away from the original model, even if the quality of the model has somehow remained high. We note that none of the quantization studies we cited and made comparisons with have included a human-study.
>
> The main challenge of user-study (apart from the direct financial cost), is that there is no clear candidate for comparison. Comparing naive quantization with SR+WSR may not be convincing enough. At the same time, few among the methods suggested by other papers offer official solutions that extend to SDXL (or beyond), and the ones we have checked are quite inflexible and are not sufficiently competitive to justify a user-study (Appendix A, G). Assuming a side-by-side user comparison is possible, what methods would the reviewer like to see compared? (and in which settings?)
>
> > WSR(p=4) increases amount of memory used for model storage by 50% relative to fp8. I would suggest to check whether mixed fp8/fp16 quantization is still inferior in terms of quality.
>
> Our main reason for picking $p=4$ is that we wanted to err on the side of caution (The 30K COCO experiment can be quite long), and consider the weight memory cost to be a relatively minor concern (please see  ``general comment’’ on this).
>
> In response to the reviews, we initiated an experiment with lower values of $p$, with positive preliminary results. For $p=2 / 4, T=50$, we measured an FID of $18.84 / 18.88$ and a PSNR of $26.15, 26.07$ respectively, which is very similar to our result for $p=4$, and can help reduce the memory requirement by an additional 750MB/ 1GB. However, we stress again that for modern architectures, most of the on-device memory is reserved for activations, which can be fully quantized when using stochastic weights and standard 8-bit GeMM kernel, regardless of the value of $p$.
>
> There is no-doubt that using mixed-precision can improve the results. However, the cost of changing between data-types is non-trivial and would be better handled by practical implementations that correspond to specific hardware accelerators.

---

> ### Author Response · Authors · 2024-11-20
> **Response 2/2 to Reviewer Xmw8 (Questions)**
>
> > The paper focuses on fp8 quantization, where performance drops relative to uncompressed model are not very pronounced. Would be interesting to see whether the conclusions are valid for the case of more aggressive quantization (say W4/A8) - i.e one should focus on reducing the bias, rather than the variance term. I suggest having at least one experiments with stronger compression to showcase the difference between RTN performance and SR.
>
> Please note that the improvements we show in FP8 quantization are significant, as indicated by the very low p-values (Figure 9). Moreover, while it is correct that these improvements are mostly in the fine details of the output image, these details are important to the users. Diffusion models are tasked with producing art. Thus, a quantized model producing an image that is 'slightly blurry’ or `mostly similar to the original image’ is typically not good enough for real-world applications. Closing the gap for lower numerical precision might be more impressive in numerical terms, but this is not the focus of this work.
>
> Specifically, for W4/A8, we were not able to get good results with 4-bit weights in our highly restrictive data-free setting. Doing so will most likely require the relaxation of some of our guiding principles, by using data for calibration and/or mixed-precision, which have significant implicit costs, as explained in our paper.
>
> > Noticeable decrease of FID metric relative to uncompressed model given high PSNR (small difference between the output of original and quantized model) looks suspicious. Do you have any explanation for this phenomenon? Do you adopt the same random seed for generation with compressed and original model? How robust is the FID value to the change of seed?
>
> We agree that this is odd, but we did not find any good explanation for this phenomenon, except attributing it to the shortcomings of FID (especially when using a refiner). The FID value changes by $\pm 1$ when using different seeds (based on limited prior experiments), but we made sure to use the same seed in all experiments (this is mandatory for the SSIM/ PSNR evaluation).
>
> > What is the sampler used for image generation? Is it the default sampler in diffusers config or DDIM?
>
> We used the default sampler (Euler) for all experiments in the main section. In appendix L, we also included results for UniPC and Heun, to measure the robustness of our method to changing hyperparameters. For Stable Diffusion 1.4 (Appendix G) we used PLMS.

---

> > ### Comment · Reviewer_Xmw8 · 2024-11-20
> > **Response**
> >
> > Thanks for your response. Some of my concerns were addressed. Therefore I decided to raise my score.

---

> > > ### Author Response · Authors · 2024-11-24
> > >
> > > We thank the reviewer for the time and effort invested in this review and the response. We are glad to have addressed some of the concerns and will be happy to hear what the remaining issues are, to help us improve our paper further.

---

### Official Review · Reviewer_76Dq · 2024-11-03

**Soundness:** 4
**Presentation:** 4
**Contribution:** 3
**Rating:** 6
**Confidence:** 5

**Summary:**

The paper presents the study of FP8 quantization (in various special forms) for diffusion models in data-free setting; it overviews various techniques to overcome quantization error/bias of aforementioned schemes, and has actionable recommendations for practical use. The presentation of the paper, including overall writing, and other evidence (proofs, plots, and supporting information) is very informative and is one of the strengths of the paper.

**Strengths:**

The paper studies the effect of fp8 quantization on diffusion models and presents various observations and recommendations for practical use. The authors convincingly show that main issue in FP8 quantization can be attributed to quantization bias and as such, propose to de-bias it with stochastic rounding schemes. Such a rounding scheme can be easily implemented for activations, but for weights it will be counter-productive; as such authors propose wight-stochastic-rounding (WSR) scheme that maintains FP8 weight copy + p-bit mantissa and certain uniform mixing scheme.  Authors  provide comparisons in terms FID/PSNR for every proposed de-biasing technique indicating that suggesting methods have merit and singling out M4E3 version of FP8 scheme as most suitable candidate for deployment.

The main strengths of the paper is thoughtfulness of the investigation and presentation: for every claim/equation in the paper there is sufficient empirical or theoretical explanation given in main paper or appendix.

**Weaknesses:**

There are only few weaknesses of the paper, and most of them were mentioned by authors in the manuscript:
1. Practical applicability of some of the proposed schemes. In the current form, only the stochastic rounding of activations can be deployed on devices; any other schemes need significant effort in software/hardware support. SR itself gives measurable improvements, but not as good as SR+WSR (based on table results)
2. Assumptions on #diffusion steps used throughout manuscript (i.e., steps > 50) are unrealistic in my opinion simply due to the nature of discussion: if we are talking about efficiency and on-device deployment, such models are most probably will be deployed in few step regime
3. p=4 for WSR scheme might be too big (in my opinion) for any practical deployment despite the shown advantage: wsr with p=4 means 12bit storage for every weight tensor, as such, there might be alternative allocation or scheme for same 12bit per weight storage. I think, for any WSR benefits, authors should try coming up with less taxing scheme on additional storage.
3. User study. I think, having user study on perceived quality difference across proposed techniques would significantly strengthen the conclusions of the paper.

**Questions:**

Please see weaknesses.

---

> ### Author Response · Authors · 2024-11-20
> **Response to Reviewer 76Dq**
>
> > Practical applicability of some of the proposed schemes. In the current form, only the stochastic rounding of activations can be deployed on devices; any other schemes need significant effort in software/hardware support. SR itself gives measurable improvements, but not as good as SR+WSR (based on table results)
>
> If the reviewer’s main concern is the reduction of on-device memory, our methods (including WSR) can be used as is to produce better quality images than any other 8-bit quantized model, since software implementation and high values of $T$ may be slow, but will not cost additional memory. Also, please see ``general comment’’.
>
> As for run-time, we do generally agree with this limitation with respect to current hardware, but believe that the main strength of the paper is the potential contribution to (near) future hardware. While no hardware implementation can be described as insignificant, Stochastic-Weights are far from being impractical: The operation’s complexity is linear with the size of weights, and can be performed preemptively with no dependency on the GeMM computation engine (e.g. Tensor Cores, in the case of Nvidia). We even made sure to use operations that have a smaller gate-count (this is why we used integer-increment, rather than addition). Modern AI accelerators include far more complex operations that offer significantly less benefit than what we proposed in this paper.
>
> > Assumptions on #diffusion steps used throughout manuscript (i.e., steps > 50) are unrealistic in my opinion simply due to the nature of discussion: if we are talking about efficiency and on-device deployment, such models are most probably will be deployed in few step regime
>
> We acknowledge that there is some contradiction between long-inference tasks and the ambition to accelerate inference. That being said, our main results for both SDXL and FLUX have shown significant benefits for 50 steps diffusion, which we consider to be very important (if not the most important) for current applications, based on the default values, tutorials, and public use-cases [1]. Higher values are indeed less practical, but they are the best solution if the main concern is on-device-memory rather than run-time, since T=200 processes require the same amount of memory as T=30 runs. Also, in the context of our work, the behaviour of long diffusion processes is informative: Equation 7, the result of our theoretical analysis, has an exponential dependency on the number of steps.
>
> Most prior works have focused on short/very short inference tasks, which we found to be a little disingenuous: they have specifically targeted easier quantization tasks with easier baselines (shorter processes have higher FID), with little to no acknowledgment of this limitation. We would also like to highlight the importance of the robustness of our method: Our networks can be used as is for shorter inference tasks, even though the benefit would be small. In contrast, existing methods, such as TensorRT, fail completely for a normal number of steps (Appendix A), even when calibrated.
>
> > p=4 for WSR scheme might be too big (in my opinion) for any practical deployment despite the shown advantage: wsr with p=4 means 12bit storage for every weight tensor, as such, there might be alternative allocation or scheme for same 12bit per weight storage. I think, for any WSR benefits, authors should try coming up with less taxing scheme on additional storage.
>
> Our main reason for picking p=4 is that we wanted to err on the side of caution (as the 30K COCO experiment), and consider the weight memory cost to be a relatively minor concern. (See: general comment)
>
> In response to the reviews, we initiated an experiment with lower values of $p$, with positive preliminary results. For $p=2 / 1, T=50$, we measured an FID of $18.84 / 18.88$ and a PSNR of $26.15, 26.07$ respectively, which is very similar to our result for $p=4$, and can help reduce the memory requirement by an additional 750MB/ 1GB. However, we stress again that for modern architectures, the main benefit of WSR is enabling 8bit$\times$8bit multiplication, and not the direct reduction in weight-memory, as we explained in the ``general comment’’.
>
> > User study. I think, having user study on perceived quality difference across proposed techniques would significantly strengthen the conclusions of the paper.
>
> The main challenge of user-study (apart from the direct cost), is that there is no clear candidate for comparison. Comparing naive quantization with SR+WSR may not be convincing enough. At the same time, few of the feasible quantization methods suggested by other papers offer official solutions that extend to SDXL (or beyond), and the ones we have checked are quite inflexible and are not sufficiently competitive to justify a user-study (Appendix A). Assuming a side-by-side user comparison is possible, what methods would the reviewer like to see compared? (and in which settings?).
>
>
> [1] https://prompthero.com

---

### Official Review · Reviewer_J51z · 2024-11-04

**Soundness:** 3
**Presentation:** 2
**Contribution:** 3
**Rating:** 6
**Confidence:** 3

**Summary:**

In this paper, the authors introduce a de-biasing approach for quantizing large-scale text-to-image diffusion models to 8 bits. They identify and mathematically analyze the bias in the existing quantization methods, which negatively impacts quantization performance, especially for long denoising sequences. To mitigate this, they use simple stochastic rounding to reduce bias. Beyond applying stochastic rounding to activations, they extend this method to weights, introducing the concept of Stochastic Weight Rounding (WSR). Experiments on SDXL and FLUX demonstrate the effectiveness of the proposed approach.

**Strengths:**

* The authors evaluate their method directly on FLUX, a state-of-the-art diffusion model, making the method practical for real-world use.
* In addition to FID, the authors report PSNR and SSIM metrics to compare their generated images with those from 16-bit models, showcasing the robustness of the proposed approach.
* The proposed method is supported by strong mathematical analysis.

**Weaknesses:**

* As noted in lines 407–417, using stochastic weights requires storing additional $p$ bits. Therefore, for a fair comparison, I suggest that the authors compare their method to W$(8+p)$A$8$ baselines. It is possible that the observed improvement primarily results from the additional weight bits.
* Writing and Presentation:
  * In Figures 1, 2, and 3, there are too many curve lines in each figure, making them difficult to interpret. Additionally, some colors are similar, and some lines overlap, making it hard to distinguish between them.
  * The authors should include visual comparisons within the main text to better showcase their results.
  * To enhance clarity, the authors should consider adding a figure that provides an overview of their main method. Currently, it is hard to grasp the key ideas from the pure text.
  * Line 131: Consider replacing "U-Net" with a different term, as FLUX uses a diffusion transformer.
  * Figure 3: The left figure lacks a label for the y-axis.
  * Line 169: The quotation marks for M$x$E$y$ are incorrect.

**Questions:**

* The concept of stochastic weights is unclear to me. Are the weights stored using $8+p$ bits while computations are still performed with 8-bit precision?
* In the paper, the authors apply their methods only to the 50-step FLUX.1-dev model. I wonder if the proposed method is also effective on the 4-step FLUX.1-schnell model.

---

> ### Author Response · Authors · 2024-11-20
> **Response to Reviewer J51z**
>
> > for a fair comparison, I suggest that the authors compare their method to W(8+p)A8 baselines. It is possible that the observed improvement primarily results from the additional weight bits.
>
> Here are the results (PSNR scores) for W(8+p)A8 quantization, next to the matching WSR scores:
>
> |          |   +0   |     +1 |     +2 |     +4 |   +1 bits |   +2 bits |   +4 bits |
> |:---------|-------:|-------:|-------:|-------:|----------:|----------:|----------:|
> | M4E3 20  | 28.228 | 28.322 | 28.232 | 27.829 |    29.078 |    29.222 |    29.286 |
> | M4E3 40  | 27.443 | 28.089 | 28.443 | 28.664 |    29.037 |    29.129 |    29.442 |
> | M4E3 60  | 27.652 | 28.48  | 28.373 | 28.481 |    28.819 |    29.483 |    29.277 |
> | M4E3 80  | 27.73  | 28.721 | 29.011 | 28.504 |    29.407 |    29.256 |    29.747 |
> | M4E3 100 | 27.726 | 28.874 | 29.84  | 29.404 |    29.767 |    29.88  |    29.709 |
> | M4E3 120 | 27.682 | 28.878 | 28.999 | 29.514 |    29.191 |    29.782 |    30.21  |
> | M4E3 140 | 27.648 | 29.25  | 29.992 | 30.288 |    29.939 |    30.265 |    31.251 |
> | M4E3 160 | 27.423 | 28.831 | 29.723 | 29.691 |    29.48  |    29.798 |    30.642 |
> | M4E3 180 | 27.069 | 28.68  | 29.344 | 29.649 |    29.168 |    30.083 |    30.91  |
> | M4E3 200 | 27.56  | 29.424 | 30.339 | 30.723 |    29.779 |    31.184 |    31.38  |
>
>
> To be clear, we did expect W(8+p)A8 quantization to be (slightly) better than stochastic rounding, since it will have the same bias as stochastic rounding while also benefiting from smaller quantization noise. The downside (in comparison with WSR) is that the W(8+p)A8 format requires more complex, unbalanced computation kernels that are less likely to exist in any hardware and would be more expensive if they were implemented. In contrast, WSR enables matrix multiplication of FP8 formats, which is already implemented in some common accelerators. See also ``general comment’’.
>
> **Presentation**
>
> We thank the reviewer for the detailed comments. We addressed most of the comments in the revised paper.
>
> A new figure with a visualized flow describing our methods was added as Figure 14. We would appreciate the reviewers’ feedback on whether it helps understand the paper.
>
> We have not added visual examples to the main paper yet and will do so according to the available space in the final revision of the paper (but will make sure to have at least a single row). The examples we will use will match the images in the supplementary materials of this paper.
>
> Improving figures 1,2 and 3 is more challenging. We agree that the information in these figures is dense, but we regard most of it as necessary (also, the other reviews have commented positively on them).
>
>
>
> **Questions**
>
> > The concept of stochastic weights is unclear to me. Are the weights stored using bits while computations are still performed with 8-bit precision?
>
> Correct. Stochastic weights essentially use the extra “p” bits for de-biasing. The computation will still be performed over 8-bit values (with higher quantization error). It works well because, according to our main argument in the paper, bias is more important than absolute error. We also leverage the fact that a common denoiser neural network has a lot of weight sharing. Consequently, the size of the activations can easily exceed the size of the weights by an order of magnitude (depending on batch size and image resolution). This means that the memory reserved for the $p$ additional bits is much smaller than the memory we “saved” by enabling 8-bit quantization for the activations. See also ``general comment’’.
>
> > In the paper, the authors apply their methods only to the 50-step FLUX.1-dev model. I wonder if the proposed method is also effective on the 4-step FLUX.1-schnell model.
>
> Based on our theoretical analysis, we don’t recommend our method for turbo settings (<10 steps). From equation 7, the advantage of our method grows exponentially with the number of steps: with T=4, we will gain little from the reduction of the bias, while paying the full toll for the added quantization noise.

---

> > ### Comment · Reviewer_J51z · 2024-11-24
> >
> > Thanks for the author response. My concerns about the paper have been addressed, and I will my rating to 6. However, I still recommend that the authors refine their writing. Additionally, I suggest that the paper include a comparison between stochastic rounding and adaptive rounding, even if only in the related work section. While adaptive rounding requires additional training, it may help reduce bias and provide valuable insights.

---

> ### Author Response · Authors · 2024-11-24
>
> We thank the reviewer for their time and feedback.
>
> We have updated the paper to discuss the differences between stochastic rounding and adaptive methods (AdaRound, AdaQuant or BREC-Q), after introducing Stochastic rounding.
>
> Please note that we have made direct [Appendix G] and indirect [Appendix A, via TensorRT] comparisons with Diffusion-Q, which uses block-wise Adaptive Rounding (Calibrated Per-Timestamp).

---

### Author Response · Authors · 2024-11-20
**Rebuttal: General Comment**

We thank the reviewers for their helpful and insightful comments.

All the reviewers acknowledged the benefits of quantizing the activations to 8bit (with SR) for its significant memory reduction. It is important to clarify that to benefit from this memory reduction in existing computation engines, both weights and activations must be quantized to a similar format. The main goal of WSR (as explained in lines 407-416) is to enable us to enjoy this benefit. Since the memory needed for the weights is much smaller in comparison to the activations memory, a direct reduction in weight memory is a secondary concern. Following the reviews, we understand that this claim may not have been sufficiently convincing and would like to add some real-world numbers to back it up.

The SDXL model has approximately 3 Billion parameters, which require a memory of 6GB for half-precision models, or 3GB for models with 8-bit weights. When running 8-bit SDXL models on H100 (80GB), A100 (40GB), or even RTX4090 (24GB), the amount of memory reserved for weights only amounts to a small percentage (3.75%, 7.5% or 12.5%, respectively) of the overall, available memory. The rest of the memory will be reserved for activations-memory, as the batch size may be increased to utilize all available resources. The weight/activation ratio is less drastic for FLUX models, but is still largely dominated by activations, at least for the commercial GPUs. (Flux-dev requires ~24GB with half-precision, and ~16GB with 8-bit quantization)

For reviewers who may still be concerned about the additional memory cost (e.g., for users running larger models or using smaller GPUs), we also added (in their individual comments, for now) results with more conservative implementations of WSR, showing that we can save around 750MB (SDXL) while maintaining the generated images’ quality. Other changes to the paper include the addition of FID-FP32 measurements (Appendix E and Figure 5d), presentation changes, and adding a new figure (Figure 14) describing our methods for an easier understanding of the compute-flow in its entirety.

---

### Meta-Review · Area_Chair_Gei4 · 2024-12-17

**Metareview:**

This paper proposes a data-free FP8 quantization method for text-to-image diffusion models, emphasizing the importance of reducing quantization bias through stochastic rounding techniques like Stochastic Weight Rounding, evaluated on models such as SDXL and FLUX.
Its strengths include addressing a relevant cost issue in diffusion models and applying to advanced models with multiple evaluation metrics. However, weaknesses are evident. The novelty of stochastic rounding is limited, and there are concerns with the evaluation setup (insufficient image numbers in some experiments) and practical implementation (hardware support and memory usage). Overall, the significant weaknesses in novelty and evaluation prevent a convincing demonstration of superiority and practicality, leading to a reject decision.

**Additional Comments On Reviewer Discussion:**

During the rebuttal period, reviewers raised several points including concerns about the number of images used in experiments for reliable assessment, the novelty of the proposed methods, the practicality of certain schemes in terms of memory and hardware implementation, and the choice of metrics for evaluation. The authors responded by adding more experimental results with a larger number of images in some cases, discussing the novelty in the context of their specific application in diffusion models, explaining the practicality aspects considering future hardware potential, and incorporating additional metrics like FID-FP32. However, in weighing these points, it was found that while some concerns were addressed to an extent, the fundamental issues regarding novelty and the overall evaluation setup still remained, leading to the decision to maintain the reject stance.

---

### Decision · Program_Chairs · 2025-01-22

Reject